# A unique sigma/anti-sigma system in the actinomycete *Actinoplanes missouriensis*

Takeaki Tezuka [1,2,4] ✉, Kyota Mitsuyama[1,4], Risa Date [1,4] &
Yasuo Ohnishi [1,3] ✉

Bacteria of the genus *Actinoplanes* form sporangia that contain dormant sporangiospores which, upon contact with water, release motile spores (zoospores) through a process called sporangium dehiscence. Here, we set out to study the molecular mechanisms behind sporangium dehiscence in *Actinoplanes missouriensis* and discover a sigma/anti-sigma system with unique features. Protein σ^SsdA contains a functional sigma factor domain and an anti-sigma factor antagonist domain, while protein SipA contains an anti-sigma factor domain and an anti-sigma factor antagonist domain. Remarkably, the two proteins interact with each other via the anti-sigma factor antagonist domain of σ^SsdA and the anti-sigma factor domain of SipA. Although it remains unclear whether the SipA/σ^SsdA system plays direct roles in sporangium dehiscence, the system seems to modulate oxidative stress responses in zoospores. In addition, we identify a two-component regulatory system (RsdK-RsdR) that represses initiation of sporangium dehiscence.

Bacterial species use a wide variety of survival strategies under conditions that are unfavourable for growth[1,2]. One strategy involves the formation of spores that are metabolically dormant and highly resilient forms of cells. While these properties provide spore-forming bacteria with an opportunity for survival over long periods, spores can sense and rapidly respond to environmental changes[3–5]. In *Bacillus subtilis*, spores can sense germinant molecules via receptor complexes embedded in their inner membranes[6–11]. After such molecules bind to their receptors, dormant spores are converted to fully active cells, mainly via two consecutive stages: germination and outgrowth[12,13]. However, few studies have focused on the maintenance of spore dormancy.

Sigma factors recognize promoter elements to initiate transcription as a component of the bacterial RNA polymerase holoenzyme. Almost all bacterial species harbour multiple sigma factors, each of which regulates the transcription of its regulon, depending on its preference for promoter sequences. Although the primary sigma factor initiates transcription of housekeeping genes under normal growth conditions, alternative sigma factors initiate transcription of a group of genes under specific conditions. To regulate the activity of alternative sigma factors, anti-sigma factors suppress their partner sigma factors via protein-protein interactions and release the partner sigma factors in response to environmental stimuli. As a mechanism for suppressing and releasing partner sigma factors, several anti-sigma factors adopt the partner-switching regulatory system, in which anti-sigma factors release cognate sigma factors by exchanging binding partners from partner sigma factors to anti-sigma factor antagonists, which are also called anti-anti-sigma factors[14–18]. The anti-sigma factor RsbW suppresses activity of *B. subtilis* σ^B under normal growth conditions. Diverse stresses lead to dephosphorylation of the anti-anti-sigma factor RsbV via activation of phosphatase activity in RsbU or RsbP. Dephosphorylated RsbV then binds tightly to RsbW to trigger the release of σ^B under adverse conditions, inducing transcription of more than 150 genes[19–22]. Conversely, in the absence of stressors, RsbW exerts kinase activity to phosphorylate RsbV and binds again to σ^B. Therefore, σ^B is modulated by the activities of the kinase RsbW and phosphatases RsbU and RsbP in this partner-switching regulatory system[14,23].

[1]Department of Biotechnology, Graduate School of Agricultural and Life Sciences, The University of Tokyo, Bunkyo-ku, Tokyo, Japan. [2]Graduate School of Infection Control Sciences, Kitasato University, Minato-ku, Tokyo, Japan. [3]Collaborative Research Institute for Innovative Microbiology, The University of Tokyo, Bunkyo-ku, Tokyo, Japan. [4]These authors contributed equally: Takeaki Tezuka, Kyota Mitsuyama, Risa Date. ✉e-mail: atezuka@mail.ecc.u-tokyo.ac.jp; ayasuo@mail.ecc.u-tokyo.ac.jp

*Actinoplanes missouriensis* is an actinomycete that grows as branched substrate mycelia during vegetative growth. It forms globose or subglobose terminal sporangia that sprout from substrate mycelia via short sporangiophores. Under laboratory conditions, *A. missouriensis* forms sporangia when cultivated on humic acid-trace element (HAT) agar. Small sporangium-like structures are produced on this agar medium after 2- or 3-day cultivation at 30 °C. Mature sporangia are then formed after incubation for 5–7 days[24–28]. Each sporangium contains a few hundred dormant spores that are encapsulated in an intrasporangial matrix called sporangium matrix. In response to water, sporangia open and release spores via a process called sporangium dehiscence[29–31]. Under laboratory conditions, sporangium dehiscence can be induced either by pouring 25 mM NH$_4$HCO$_3$ onto sporangia formed on HAT agar or suspending the sporangia harvested from the agar surface in 25 mM histidine solution and incubating for 1 h. Under the latter conditions, the sporangia appear phase-bright immediately after suspension and then the sporangium membrane gradually becomes transparent before spore release when observed by phase-contrast microscopy (see Fig. 1a–c). After release from sporangia, spores swim in aquatic environments using flagella as zoospores and exhibit chemotactic properties. In niches suitable for vegetative growth, zoospores stop swimming and resume vegetative growth[32,33].

According to this complex life cycle, the entire revival process, from dormant sporangiospores to vegetative growth of mycelia in *A. missouriensis*, can be divided into three stages: (i) activation of dormant sporangiospores through contact with water, followed by sporangium dehiscence and spore release; (ii) swimming of motile zoospores in aquatic environments; and (iii) emergence of germ tubes in favourable niches[28,33]. In this process, sporangium dehiscence can be considered as the initial stage of awakening of dormant sporangiospores in response to environmental stimuli. Although many bacterial species, including clinically important pathogens, form spores, insights into the mechanism of their awakening process induced by environmental stimuli are limited, and molecular details of signal transduction in this process are not yet clear[6,7,13].

In this study, we aimed to elucidate the molecular mechanisms underlying sporangiospore activation in *A. missouriensis* by isolating and analyzing mutant strains that produce sporangia defective in sporangium dehiscence to release spores. We identified a protein pair composed of the sigma factor σ$^{SsdA}$ (AMIS_54240) and its cognate anti-sigma factor SipA (AMIS_54230), which is involved in oxidative stress responses in zoospores. Functional analysis of this regulatory system also identified a two-component regulatory system, RsdK-RsdR, as a key factor that enhances resistance to sporangium dehiscence and delays the initiation of sporangiospore awakening.

## Results
### SipA is crucial for sporangium dehiscence

Because *A. missouriensis* sporangiospores are enclosed by a sporangium matrix and membrane, we hypothesized that these protective

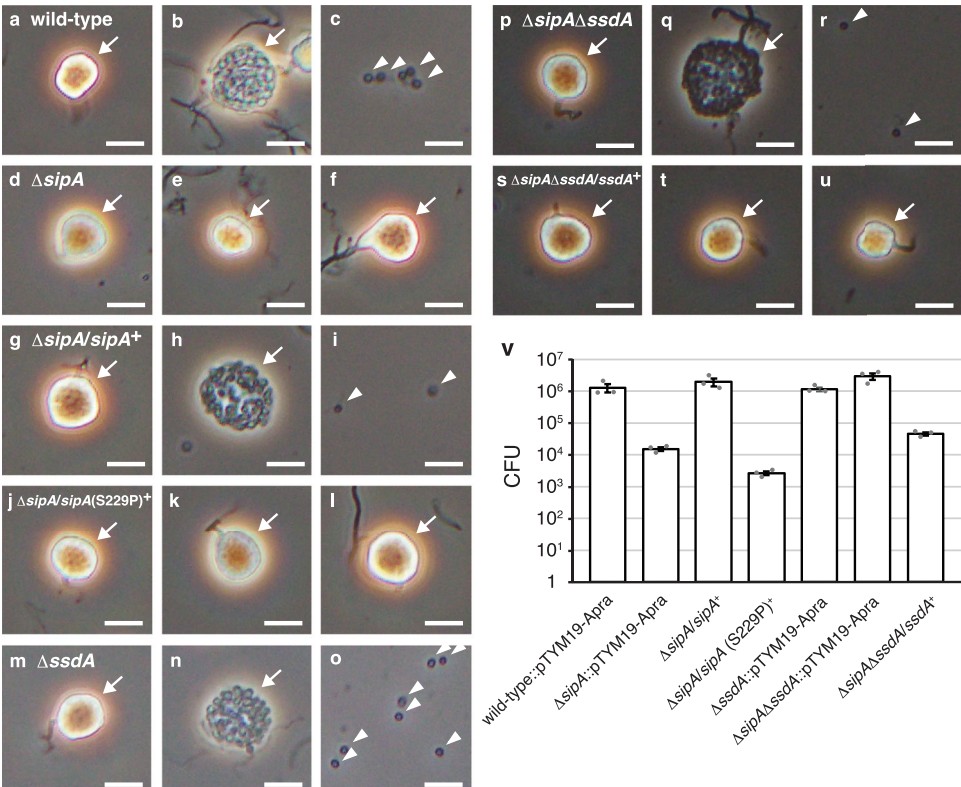

**Fig. 1 | Observation of sporangium dehiscence and number of spores released from sporangia. a–u** Observation of sporangia and zoospores using phase-contrast microscopy. Sporangia produced on HAT agar were harvested and suspended in 25 mM histidine solution to induce sporangium dehiscence. Microscopic images of the wild-type strain (**a–c**), ΔsipA strain (**d–f**), ΔsipA strain harbouring *sipA* complementation plasmid (**g–i**), ΔsipA strain harbouring *sipA* (S229P)-expressing plasmid (**j–l**), ΔssdA strain (**m–o**), ΔsipAΔssdA strain (**p–r**), and ΔsipAΔssdA strain harbouring *ssdA* complementation plasmid (**s–u**). Images in **a, d, g, j, m, p, s** were obtained immediately after suspension. Images in **b, e, h, k, n, q, t** were obtained 15 min after suspension. Images in **c, f, i, l, o, r, u** were obtained 30 min after suspension.

Immediately after suspension, the sporangia appeared phase-bright (**a, d, g, j, m, p, s**). The sporangium membranes gradually became transparent before spore release (**b, h, n, q**). Sporangia (including those whose membrane became transparent) and released spores are indicated by arrows and arrowheads, respectively. Scale bars, 5 μm. The entire images of each microscopic field are shown in Fig. S4. **v** Number of spores released from the sporangia. Each strain was cultivated on HAT agar at 30 °C for 7 days. Zoospores released from the sporangia formed on one HAT agar plate by pouring 25 mM NH$_4$HCO$_3$ solution were counted as colony forming unit (CFU) on YBNM agar. The values represent the mean ± standard error of three biological replicates. Source data are provided as a Source Data file.

layers act as a shield between the sporangiospores and their surroundings, enabling them to confront environmental stresses. To investigate the heat tolerance of sporangiospores and zoospores, we suspended the sporangia and zoospores of the wild-type strain separately in distilled water and incubated them at 50 °C for 90 min. As expected, the survival rates of sporangiospores (spores in a sporangium), determined every 30 min after heat treatment, were much higher than those of zoospores (spores released from sporangia) (Fig. S1). Based on this apparent heat tolerance of sporangiospores, we developed a scheme for mutant screening in which sporangium dehiscence-deficient strains were enriched from a mutant library generated via UV irradiation of wild-type zoospores. Briefly, (i) sporangia harvested from the surface of HAT agar were suspended and incubated for 1 h in 25 mM histidine solution to induce sporangium dehiscence. Next, (ii) the solution was incubated at 50°C for 30 min to enrich the sporangia that did not open under these conditions, followed by cultivation on HAT agar for sporangium formation. After repeating procedures (i) and (ii), we isolated mutant strains by isolating single colonies by cultivating a portion of the heat-treated solution on yeast extract-beef extract-NZ amine-maltose monohydrate (YBNM) agar. We then cultivated each strain on HAT agar for sporangium formation and examined sporangium dehiscence using a phase-contrast microscope by suspending and incubating the sporangia harvested from the agar surface in a dehiscence-inducing solution. Consequently, we identified 27 mutants, designated as M-1 to M-27, whose sporangia were not fully mature to release spores or were defective only in dehiscence. To identify the generated mutations, we determined the genome sequences of all the mutant strains (Table S1).

Of the 27 sequenced strains, we found that 12 strains (M-2 to M-10, M-16, M-18, and M-25) carried mutations in *hhkA*, which encodes a hybrid sensor histidine kinase[34], and that two strains (M-1 and M-26) carried mutations in *tcrA*, which encodes a response regulator[35]. We previously reported that HhkA and TcrA probably comprise a cognate two-component system that controls the transcription of a number of genes involved in sporangium formation, spore dormancy, and sporangium dehiscence[34,35]. Therefore, mutations in *hhkA* and *tcrA* are likely to explain the deficiency in sporangium dehiscence observed in these mutants. Thus, we excluded these mutants from further analysis. Furthermore, we excluded mutants M-15 and M-17 because they carried comparably many mutations (at 10 and 8 loci, respectively). As a result, we focused on the remaining 11 strains and quantified spores released from the sporangia. The number of spores released from sporangia was remarkably decreased in the M-12, M-19, and M-20 mutants compared to that in the wild-type strain (Fig. S2). Hereafter, we focused on the mutant M-12 because only a single-nucleotide variant within *AMIS_54230* was identified, which replaced Ser-229 with Pro in the 243 amino acid product (Table S1). We designated AMIS_54230 as SipA (SsdA inhibitor protein). A protein database search using the Conserved Domain Database (https://www.ncbi.nlm.nih.gov/Structure/cdd/cdd.shtml) suggested that SipA possesses an STAS domain (cl00604; residues 6–103) and histidine kinase-like ATPase domain of RsbW (cd16936; residues 158–239). Previous studies have reported that the STAS domain is located in sulfate transporters and in anti-sigma factor antagonists[36,37]. Thus, this in silico search predicted that SipA harbours both anti-anti-sigma factor and anti-sigma factor domains in its N- and C-terminal portions, respectively (see Fig. 2d).

To examine the in vivo function of SipA, we generated a *sipA* null mutant (Δ*sipA*) strain. Because no difference was observed between the wild-type and Δ*sipA* strains by macroscopic observation of mycelia or sporangia formed on YBNM and HAT agar (data not shown), we examined sporangium formation in detail by observing mycelia and sporangia grown on HAT agar using scanning electron microscopy (SEM). However, both the strains produced normal sporangia under the tested conditions (Fig. S3a, b). Next, we examined sporangium dehiscence and motility of zoospores using phase-contrast

microscopy. Sporangium dehiscence was severely repressed in mutant Δ*sipA*, and zoospores were scarcely observed (Fig. 1a–f; Fig. S4a–f). Hence, we quantified the spores released from sporangia in the wild-type and Δ*sipA* strains, both of which contained the chromosome-integrating vector pTYM19-Apra (empty vector), by counting the colonies formed on YBNM agar after incubation at 30 °C for 2 days. Because the solution containing the zoospores released from sporangia was filtered through a 5-μm membrane filter to eliminate hyphae and sporangia in this experiment (see Methods), all colonies formed on YBNM agar came from the spores released from sporangia[35]. While wild-type sporangia formed on a single HAT agar plate released over $10^6$ spores, Δ*sipA* sporangia released only approximately $10^4$ spores per plate (Fig. 1v), which is consistent with the phase-contrast microscopy observations (Fig. 1a–f). In a gene complementation test, the introduction of pTYM19-Apra carrying *sipA* into the Δ*sipA* strain resulted in complete restoration of sporangium dehiscence and the number of spores released from sporangia (Fig. 1g–i, v; Fig. S4g–i). As described above, Ser-229 in SipA is replaced with Pro in mutant M-12. Therefore, we generated and introduced mutated *sipA* encoding SipA (S229P) into the Δ*sipA* strain in parallel. Sporangium dehiscence and the number of spores released from the sporangia were not restored by the introduction of this mutated gene (Fig. 1j–l, v; Fig. S4j–l), indicating that the S229P mutation renders SipA non-functional with respect to these phenotypic changes. These results clearly demonstrate that SipA is crucial for sporangium dehiscence.

## σ$^{SsdA}$ is responsible for the deficiency in sporangium dehiscence in the Δ*sipA* strain

Since SipA possesses an anti-sigma factor domain in its C-terminal portion, we postulated that SipA controls sporangium dehiscence by interacting with a target sigma factor. To identify the target of SipA, we isolated suppressor mutants from a mutant library generated by UV irradiation of zoospores of mutant M-12. For this screening, (i) sporangia harvested from HAT agar were suspended and incubated in 25 mM histidine solution to induce sporangium dehiscence, and (ii) zoospores released from sporangia were enriched by filtering the solution through a 5-μm membrane filter to eliminate hyphae and sporangia, followed by cultivation on HAT agar for sporangium formation. After repeating procedures (i) and (ii), we isolated mutant strains by isolating single colonies by cultivating a portion of the zoospore-containing filtrate on YBNM agar. We then cultivated each strain on HAT agar for sporangium formation and examined sporangium dehiscence in 25 mM histidine solution. Consequently, we obtained five suppressor mutants named S-7, S-9, S-14, S-20, and S-21, whose sporangia opened normally and released spores (data not shown). To identify the generated mutations, we determined the genome sequences of the five strains (Table S2). Considering the obtained data, we focused on *AMIS_54240*, which is located next to *sipA*, because single-nucleotide variants within this gene were identified in four of the five strains at different positions (Table S2). In particular, only one single-nucleotide variant, in which Ser-100 is replaced with Pro in the 379 amino acid product, was identified in strain S-14 (Table S2). We designated AMIS_54240 as SsdA (sigma factor presumably involved in spore dormancy). A protein database search using the Conserved Domain Database indicated that σ$^{SsdA}$ possesses a sigma factor domain of sigma-B/F/G (cl37200; residues 37–260) and an STAS domain (residues 282–377), suggesting that σ$^{SsdA}$ functions as a sigma factor containing an additional anti-sigma factor antagonist domain in its C-terminal region (see Fig. 2d).

To examine the in vivo function of σ$^{SsdA}$, we generated an *ssdA* null mutant (Δ*ssdA*) strain by deleting most of its coding sequence from the wild-type strain. No difference in macroscopically observed mycelia or sporangia formed on YBNM and HAT agar was found in the Δ*ssdA* strain compared to the wild-type strain (data not shown). We also examined mycelia and sporangia grown on HAT agar using SEM, but

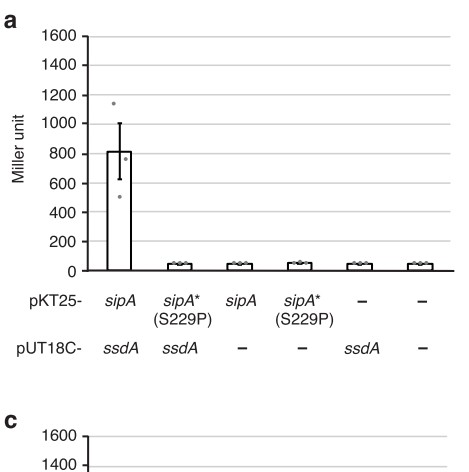

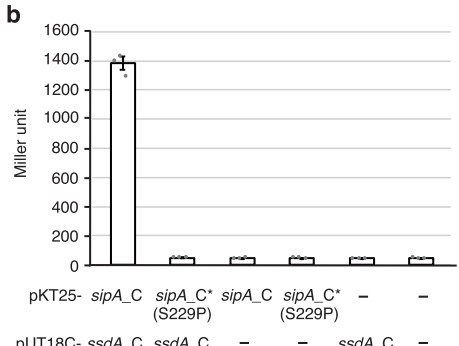

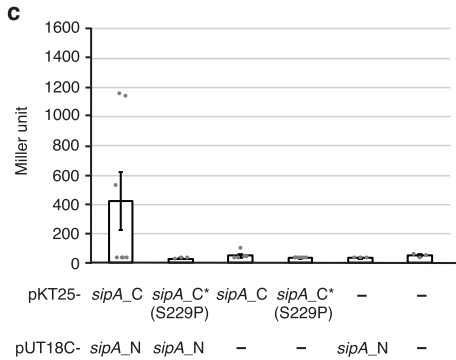

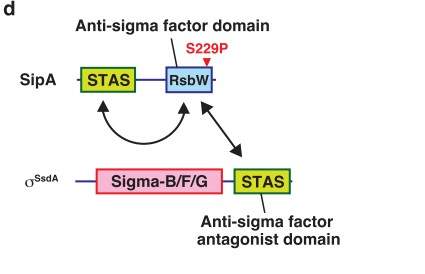

**Fig. 2 | BACTH assays for SipA and σ^SsdA. a–c** β-Galactosidase activity (Miller unit) of *E. coli* BTH101 co-transformed with the following two plasmids: plasmids harbouring *sipA* and *ssdA* individually (**a**); plasmids harbouring *sipA*_C and *ssdA*_C genes encoding the C-terminal anti-sigma factor domain of SipA and the C-terminal anti-sigma factor antagonist domain of σ^SsdA, respectively (**b**); and plasmids harbouring the *sipA*_N and *sipA*_C genes encoding the N-terminal anti-sigma factor antagonist and C-terminal anti-sigma factor domains of SipA, respectively (**c**). In **a–c**, the plasmid for production of the SipA (S229P) variant was also used. Mutated genes are shown with asterisks. Empty vectors expressing only T18 and T25 domains of adenylate cyclase were used as vector controls. Data are means ± standard error. In **c**, seven, five, and six biologically independent samples were analyzed for the transformants with pKT25-*sipA*_C and pUT18C-*sipA*_N, with pKT25-*sipA*_C and pUT18C, and with pKT25-*sipA*_C (S229P) and pUT18C, respectively. For the remaining transformants, three biologically independent samples were analyzed. Source data are provided as a Source Data file. **d** A schematic diagram of domain structures of SipA and σ^SsdA. Domain combinations whose interactions were detected in the BACTH assays are indicated by double-headed arrows. Location of S229P replacement in SipA is indicated by a red arrowhead.

the Δ*ssdA* strain produced normal sporangia (Fig. S3c). Furthermore, the Δ*ssdA* sporangia opened normally and released spores under dehiscence-inducing conditions (Fig. 1m–o; Fig. S4m–o). Sporangia of the Δ*ssdA* strain, which contained pTYM19-Apra, also released a similar number of spores as wild-type sporangia (Fig. 1v). We then assumed that the function of σ^SsdA might be responsible for the long-term survival of sporangiospores. Thus, we hypothesized that if dormancy continues for a long time, the sporangiospores of mutant Δ*ssdA* would encounter some problems, which would reduce the germination rate. However, approximately 2-month-old sporangia of the Δ*ssdA* strain showed no decrease in the number of spores released by sporangium dehiscence, similar to that of the wild-type strain (Fig. S5). These observations indicated that the deletion of *ssdA* had no effect on sporangium formation and dehiscence under the tested conditions.

Considering that the sporangia of strain S-14, which carried mutations in both *sipA* and *ssdA*, opened normally under dehiscence-inducing conditions, we generated a double mutant of *sipA* and *ssdA* (Δ*sipA*Δ*ssdA*) by deleting most of the coding sequence of *ssdA* using the Δ*sipA* strain as the parental strain. No differences were observed in the mycelia or sporangia formed on YBNM and HAT agar between the wild-type and Δ*sipA*Δ*ssdA* strains (data not shown). The Δ*sipA*Δ*ssdA* strain also formed normal sporangia, similar to those of the wild-type strain as observed using SEM (Fig. S3d). Hence, we analyzed sporangium dehiscence of the double mutant by phase-contrast microscopy. As anticipated, the sporangia of the Δ*sipA*Δ*ssdA* strain opened normally (Fig. 1p–r; Fig. S4p–r), which was in marked contrast to that of the Δ*sipA* strain, whose sporangia barely opened (Fig. 1d–f). Consistent

with this observation, the number of spores released from the sporangia of the Δ*sipA*Δ*ssdA* strain, which contained pTYM19-Apra, was similar to that of wild-type sporangia (Fig. 1v). In a gene complementation test, introduction of pTYM19-Apra carrying *ssdA* into the Δ*sipA*Δ*ssdA* strain resulted in the loss of sporangium dehiscence and decrease in the number of spores released from the sporangia (Fig. 1s–u, v; Fig. S4s–u). These results clearly demonstrate that σ^SsdA is responsible for the deficiency in sporangium dehiscence in the Δ*sipA* strain.

## Anti-sigma factor domain of SipA interacts with anti-sigma factor antagonist domains of SipA and σ^SsdA

The phenotypic investigations described above suggest that SipA functions as an anti-sigma factor for σ^SsdA via protein-protein interactions. To verify this possibility, we performed a bacterial adenylate cyclase-based two-hybrid (BACTH) assay using *Escherichia coli* as a host. We detected a significant increase in β-galactosidase activity in transformants containing both *sipA*- and *ssdA*-expressing plasmids compared to transformants with empty vectors, which indicated a direct interaction between SipA and σ^SsdA (Fig. 2a). As described above, the SipA (S229P) variant was not functional in the gene complementation test (Fig. 1j–l, v). Consequently, we analyzed whether this mutant protein interacted with σ^SsdA. We did not detect any interactions between SipA (S229P) and σ^SsdA (Fig. 2a), supporting the notion that the SipA (S229P) variant is not functional in *A. missouriensis* because of its inability to repress σ^SsdA function. There are two possible explanations for the negative effect of S229P replacement on the

function of SipA: (i) Ser-229 plays a significant role in the interaction between SipA and σ$^{SsdA}$, and the SipA (S229P) variant cannot bind to σ$^{SsdA}$; and (ii) S229P replacement considerably decreases the stability of SipA, and the structurally unstable SipA (S229P) variant cannot exert its function. Based on the predicted structure of the SipA−σ$^{SsdA}$ complex, we believe that the latter explanation is more probable (see below). Meanwhile, we also tested whether SipA interacted with σ$^{SsdA}$ (S100P) variant produced in strain S-14 (Table S2); a significant increase in β-galactosidase activity was detected, indicating that SipA interacts with σ$^{SsdA}$ (S100P) (Fig. S6a, lane 3). In contrast, no interaction was detected between SipA (S229P) and σ$^{SsdA}$ (S100P) (Fig. S6a, lane 4), suggesting that normal sporangium dehiscence observed in strain S-14 was not due to an interaction between these mutant proteins. Ser-100 is located in Region 2 of the sigma factor domain of σ$^{SsdA}$ (Fig. S5b), which is involved in the interaction between the sigma factor and −10 element of its target promoters[38]. Thus, we speculate that σ$^{SsdA}$ (S100P) is inactive because it fails to recognize its target promoters.

As described above, SipA possesses both anti-sigma factor and anti-sigma factor antagonist domains in its C- and N-terminal portions, respectively, whereas σ$^{SsdA}$ carries an anti-sigma factor antagonist domain in its C-terminal portion (Fig. 2d). To further characterize the interaction between SipA and σ$^{SsdA}$, we generated plasmids for the production of each of these domains and used them in the BACTH assay. First, we showed that the C-terminal anti-sigma factor domain, but not the N-terminal anti-sigma factor antagonist domain, of SipA interacted with full-length σ$^{SsdA}$ (Fig. S6a, lanes 5 and 6). In this truncated form, the S229P replacement in SipA also abolished its interaction with σ$^{SsdA}$ (Fig. S6a, lane 7), supporting the hypothesis that the anti-sigma factor domain of SipA is responsible for its interaction with σ$^{SsdA}$. Because anti-sigma factor RsbW interacts with its cognate sigma factor σ$^B$ to inhibit its activity in *B. subtilis*[14], we hypothesized that the anti-sigma factor domain of SipA would bind to the sigma factor domain of σ$^{SsdA}$. Therefore, we generated an *E. coli* transformant that produced the sigma factor domain of σ$^{SsdA}$ together with full-length SipA. However, no interaction between SipA and the sigma factor domain of σ$^{SsdA}$ was detected (Fig. S6a, lane 8). In contrast, interaction between SipA and the anti-sigma factor antagonist domain of σ$^{SsdA}$ was detected (Fig. S6a, lane 9). These results suggest that SipA and σ$^{SsdA}$ interact via binding between the anti-sigma factor domain of SipA and the anti-sigma factor antagonist domain of σ$^{SsdA}$ (Fig. 2d). A significant increase in β-galactosidase activity was detected in the transformant producing the anti-sigma factor domain of SipA and anti-sigma factor antagonist domain of σ$^{SsdA}$ compared to the transformant producing the anti-sigma factor domain of SipA and sigma factor domain of σ$^{SsdA}$ (Fig. 2b and Fig. S6a, lanes 10 and 12), which was comparable to that in the strain co-transformed with empty vectors (Fig. S6a, lane 24). The S229P mutation in SipA also nullified the interaction between the anti-sigma factor domain of SipA and anti-sigma factor antagonist domain of σ$^{SsdA}$ (Fig. 2b). Furthermore, we focused on the replacement of L372P in σ$^{SsdA}$, which was identified in strain S-20 (Table S2), because this amino acid is located in the STAS domain (Fig. S6b). Notably, the remaining three mutations, D68N, F91S, and S100P, identified in suppressor strains S-21, S-9, and S-14, respectively, are located at the N-terminal sigma factor domain (Table S2). We hypothesized that the SipA (S229P) variant interacts with σ$^{SsdA}$ (L372P) variant, leading to normal sporangium dehiscence in strain S-20. Therefore, we generated an *E. coli* transformant that produced both SipA and σ$^{SsdA}$ variants. However, no interaction between SipA (S229P) and σ$^{SsdA}$ (L372P) was detected (Fig. S6a, lane 13), suggesting that the L372P replacement may also inhibit sigma factor function of σ$^{SsdA}$.

During these BACTH assays, we noticed that the β-galactosidase activity of the transformant producing the anti-sigma factor domain of SipA and the anti-sigma factor antagonist domain of σ$^{SsdA}$ was remarkably higher than that of the transformant producing full-length SipA and the anti-sigma factor antagonist domain of σ$^{SsdA}$ (Fig. S6a,

lanes 9 and 10), suggesting that the N-terminal anti-sigma factor antagonist domain of SipA hinders the interaction between SipA and σ$^{SsdA}$. Thus, we performed an additional BACTH assay, in which a possible interaction between the anti-sigma factor and anti-sigma factor antagonist domains of SipA was examined. As anticipated, interaction between the two domains of SipA was detected (Fig. 2c). In addition, S229P replacement in anti-sigma factor domain also abolished this interaction (Fig. 2c). Collectively, these data clearly demonstrate that the anti-sigma factor domain of SipA interacts with anti-sigma factor antagonist domains of SipA and σ$^{SsdA}$ (Fig. 2d).

## AlphaFold-Multimer predicted different binding manners of the anti-sigma factor domain of SipA to the anti-sigma factor antagonist domains of SipA and σ$^{SsdA}$

We predicted a heterodimeric structure of the full-length SipA and σ$^{SsdA}$ proteins using the protein structure prediction tool AlphaFold-Multimer ColabFold version[39]. AlphaFold-Multimer generated a three-dimensional model in which the anti-sigma factor domain of SipA was located between two anti-sigma factor antagonist domains of SipA and σ$^{SsdA}$ with high accuracy (Fig. 3, Fig. S7a, Supplementary Data 1). Since we expected the anti-sigma factor domain of SipA to bind to the anti-sigma factor antagonist domains of SipA and σ$^{SsdA}$ in the same manner, we further predicted the following structures (Supplementary Data 1): (i) full-length SipA (by AlphaFold; Fig. S7d); (ii) a complex composed of separate anti-sigma factor and anti-sigma factor antagonist domains of SipA (Fig. S7g); (iii) a heterodimer complex composed of the anti-sigma factor domain of SipA and the anti-sigma factor antagonist domain of σ$^{SsdA}$ (Fig. S7j); and (iv) a heterodimer complex composed of full-length SipA and the anti-sigma factor antagonist domain of σ$^{SsdA}$ (Fig. S7m). All predicted structures showed similar configurations with high accuracy, indicating that the anti-sigma factor domain of SipA interacts with anti-sigma factor antagonist domains of SipA and σ$^{SsdA}$ in different manners; it binds to each of the two anti-sigma factor antagonist domains on opposite faces. Because our BACTH assay suggested that the anti-sigma factor antagonist domains of SipA and

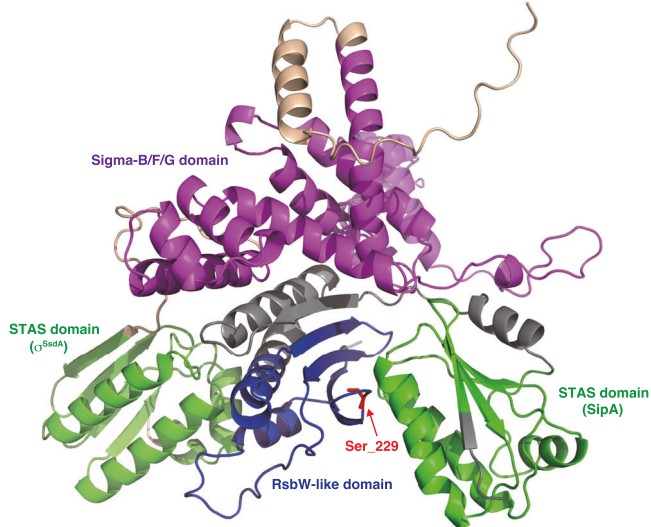

**Fig. 3 | AlphaFold-Multimer-based prediction of the SipA-σ$^{SsdA}$ complex structure.** Polypeptides are shown by ribbon representation and coloured green for the STAS domains (SipA and σ$^{SsdA}$), blue for the RsbW-like domain (SipA), and magenta for the sigma-B/F/G domain (σ$^{SsdA}$). The remaining residues of SipA and σ$^{SsdA}$ are coloured grey and pale orange, respectively. Ser-229 in SipA is indicated by a red arrow. PDB files for predicted structures are available in Supplementary Data 1. Predicted local distance difference test (pLDDT) and predicted aligned error (PAE) scores are shown in Fig. S7.

σ<sup>SsdA</sup> compete for binding to the anti-sigma factor domain of SipA, this result was unexpected. However, we believe that the predicted structures are reliable and speculate that the result of the BACTH assay, in which the presence of the N-terminal anti-sigma factor antagonist domain of SipA lowered β-galactosidase activity triggered by the interaction between SipA and σ<sup>SsdA</sup>, arises from the difference in efficiency of the formation of active adenylate cyclase from two fusion proteins.

Ser-229 in SipA was predicted to be located at the interface between the anti-sigma factor and anti-sigma factor antagonist domains of SipA (Fig. 3, Fig. S7a, d, g, m). This predicted structure supports the experimental result that S229P replacement abolished the interaction between these two domains of SipA (Fig. 2c). However, it is difficult for this predicted structure to explain why the S229P mutation abolished the interaction between the anti-sigma factor domain of SipA and the anti-sigma factor antagonist domain of σ<sup>SsdA</sup>. We speculate that S229P replacement increases the instability of SipA and abolishes the formation of the SipA–σ<sup>SsdA</sup> complex.

## Transcription of *sipA* and *ssdA* occurs during sporangium formation and dehiscence, suggesting a possible function of the SipA–σ<sup>SsdA</sup> pair

Considering the results of phenotypic investigations (Fig. 1), the SipA–σ<sup>SsdA</sup> pair seems to regulate the genes involved in sporangium dehiscence; σ<sup>SsdA</sup> apparently repressed sporangium dehiscence, and SipA appeared to neutralize the negative effect of σ<sup>SsdA</sup> on sporangium dehiscence. Notably, neither *sipA* nor *ssdA* appears to be necessary for sporangium formation (Fig. S3b, c). To examine the time points at which transcription of these genes occurs, we performed exhaustive transcriptome analysis at various time points during the life cycle of *A. missouriensis* using RNA sequencing (RNA-Seq) (accession No. DRA012687 in the DDBJ Sequence Read Archive). For sporangium formation, RNA samples were prepared from the mycelia and/or sporangia grown on HAT agar for 1, 3, 6, and 15 days in triplicate at each time point. For sporangium dehiscence, we prepared RNA samples from sporangia (including some substrate hyphae) suspended and incubated in 25 mM histidine solution for 0, 15, and 60 min to induce sporangium dehiscence in triplicate at each time point. Transcription of *sipA* and *ssdA* occurred during sporangium formation, as well as during sporangium dehiscence (Fig. S8a). We visualized two scenarios for the function of the SipA–σ<sup>SsdA</sup> pair. In Scenario I, it functions during sporangium dehiscence as an important switch to determine whether to awaken or continue dormancy. In Scenario II, it functions during and/or after sporangium formation to produce and/or retain "normal" sporangia that are resistant to opening when they are placed under dehiscence-inducing conditions. Scenario I is simple and attractive, but has a serious flaw. When cells decide to awaken, σ<sup>SsdA</sup> function should be impaired by SipA. In other words, shut-off of gene expression is the transition point; however, this process is very inefficient. Therefore, we postulated Scenario II as a probable scenario and assumed that sporangia that are apparently normal but highly resistant to opening are produced by the Δ*sipA* mutant, where σ<sup>SsdA</sup> exerts its functions more efficiently than usual because of the absence of SipA during and/or after sporangium formation. We also assumed that the function of σ<sup>SsdA</sup> might be analogous to that of *E. coli* σ<sup>S</sup>, a stationary phase-specific sigma factor responsible for long-term survival under nutrient-deficient conditions[40,41].

## Genes regulated by σ<sup>SsdA</sup>

According to the above-mentioned assumption, we hypothesized that σ<sup>SsdA</sup> activates genes required for maintaining spore dormancy in the sporangium. To define the σ<sup>SsdA</sup> regulon, we compared the transcriptomes of mutant Δ*sipA* ("hyper-active" σ<sup>SsdA</sup> strain), in which σ<sup>SsdA</sup> activity is expected to be enhanced, and mutant Δ*sipA*Δ*ssdA* (no σ<sup>SsdA</sup> strain) using RNA-Seq. Total RNA was extracted from mixtures of vegetative hyphae and sporangia grown on HAT agar at 30 °C for 6 days

in triplicate for each strain, because σ<sup>SsdA</sup> is speculated to activate its regulon in mature sporangia to maintain spore dormancy. From the data obtained by RNA-Seq, we extracted genes that met the following criteria as up- and down-regulated genes in the Δ*sipA* strain compared to Δ*sipA*Δ*ssdA* strain: (i) average number of reads per kilobase of coding sequence per million mapped reads (RPKM) in the Δ*sipA* strain more than 2.0-fold (upregulation) or less than 0.5-fold (downregulation) compared to the Δ*sipA*Δ*ssdA* strain and (ii) statistical *q* values less than 0.05. Surprisingly, transcriptional profiles of both the strains were significantly different, although the sporangia produced showed no differences in appearance; the transcript levels of 546 genes were significantly changed, with 213 and 333 genes being up- and downregulated, respectively, in the Δ*sipA* strain (Fig. 4a; Supplementary Data 2 and 3). This result strongly indicated the importance of the SipA–σ<sup>SsdA</sup> pair in the late stage of (or even after) sporangium formation, although the difference may constitute many indirect effects of the absence of σ<sup>SsdA</sup> and/or the presence of "hyper-active" σ<sup>SsdA</sup>. We assumed that the direct target of σ<sup>SsdA</sup> would be limited to a small number of genes among the 213 genes upregulated in the Δ*sipA* strain.

Therefore, we searched for conserved sequence motifs within the regions upstream of the genes upregulated in the Δ*sipA* strain to identify σ<sup>SsdA</sup>-recognizing promoters. A computational search using the MEME algorithm (http://meme-suite.org/tools/meme) produced a highly enriched sequence motif: 5′-GnTT-n₁₄-CGGGTA-3′. Although the spacer length between the −10 and −35 elements of this motif is shorter than that of the target promoters of the principal sigma factor, a similar spacer length (14–16 bp) has been reported for the target promoters of σ<sup>B</sup> in *Streptomyces coelicolor* A3(2)[42]. To exhaustively define the σ<sup>SsdA</sup> regulon, we thoroughly investigated the upstream regions of the 213 genes upregulated in mutant Δ*sipA* for this conserved motif using the FIMO algorithm (http://meme-suite.org/tools/fimo). Consequently, 17 regions were found to contain sequences similar to the conserved motif at appropriate positions with respect to the predicted transcriptional start points (Fig. 4b and Fig. S9). To demonstrate that σ<sup>SsdA</sup> recognizes these sequences as promoters, we produced and purified a recombinant σ<sup>SsdA</sup> protein with a polyhistidine tag at its N-terminus (His-σ<sup>SsdA</sup>), using *E. coli* as a host (Fig. 4c). We also prepared two DNA templates containing upstream regions of *AMIS_25220* and *AMIS_68780* (Fig. 4d and Fig. S9) and used these for an in vitro transcription assay employing His-σ<sup>SsdA</sup>. In this assay, we used a sigma factor-free RNA polymerase core complex from *E. coli* because using this core complex sigma factors can recognize their target promoters[43]. Signals corresponding to the transcripts from transcriptional start points were detected in the assays using both the templates (Fig. 4d), demonstrating that the sequence motifs are σ<sup>SsdA</sup>-recognizing promoters.

Next, we examined transcription profiles of 17 σ<sup>SsdA</sup>-dependent genes. As shown in Fig. S8b–d, although the transcription levels differed from gene to gene, their transcription profiles were almost similar; they were transcribed during sporangium formation, as well as during sporangium dehiscence, similar to *sipA* and *ssdA* (Fig. S8a). This result also contradicts Scenario I; the SipA–σ<sup>SsdA</sup> pair does not seem to function as a switch to determine the timing of sporangium dehiscence because the transcription levels of its target genes did not drastically change before and after sporangium dehiscence (Fig. S8b–d, D₀ and D₆₀). We speculated that SipA modulates the function of σ<sup>SsdA</sup> to adjust the transcription levels of its target genes to appropriate levels in the environment. Meanwhile, the transcription profiles slightly differed for several genes. For instance, the transcript level of *AMIS_25220* decreased during sporangium dehiscence, whereas that of *AMIS_68780* increased. Unknown transcriptional regulators may contribute to the differences of the transcript levels of these genes.

Of the 17 probable σ<sup>SsdA</sup>-target genes, 11 encode hypothetical proteins. The remaining six genes encode five putative enzymes: short chain dehydrogenase (AMIS_11150), catalase (AMIS_37880), glutamate-

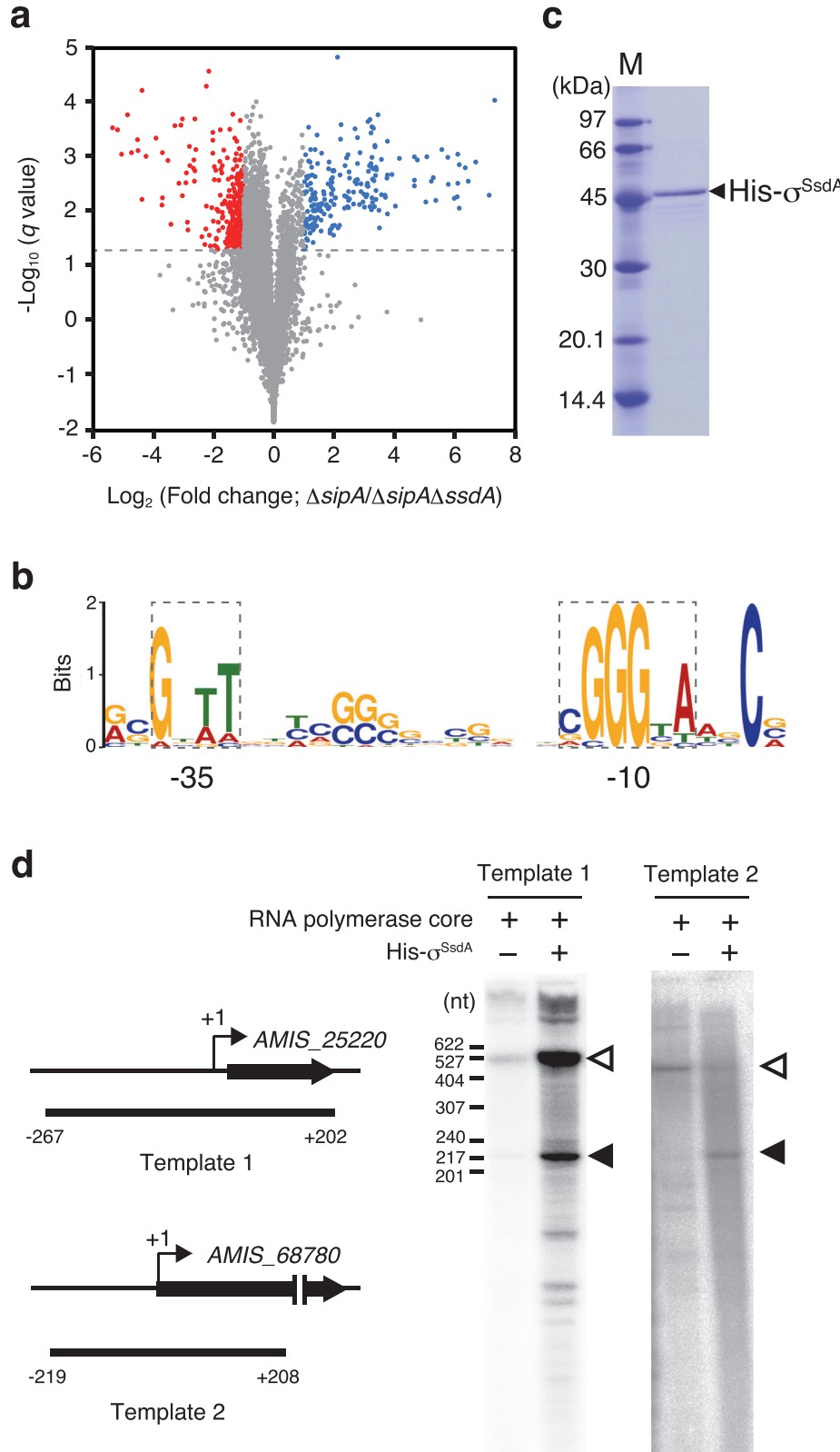

cysteine ligase (AMIS_39380), luciferase-like monooxygenase (AMIS_63160), and alcohol dehydrogenase (AMIS_40720), and one putative two-component system response regulator (AMIS_4840). Thus, this putative regulator (upregulated 54.8-fold in mutant ΔsipA) may be a key factor affecting the transcription of other up- and down-regulated genes in mutant ΔsipA.

## Identification of a two-component regulatory system (RsdK-RsdR) as a determinant of deficiency in sporangium dehiscence in mutant ΔsipA

As described above, we successfully isolated five suppressor strains whose sporangia open normally under dehiscence-inducing conditions (Table S2). Among these strains, four strains (S-9, S-14, S-20, and

**Fig. 4 | RNA-Seq analysis and in vitro transcription using recombinant σ^SsdA.** **a** Volcano plot of differential expression. Each gene was plotted based on fold-change in the Δ*sipA* strain versus Δ*sipA*Δ*ssdA* strain and the *q* value. Genes differentially expressed in Δ*sipA* strain compared to Δ*sipA*Δ*ssdA* strain are highlighted by colour: blue and red dots indicate up- and down-regulated (>2.0-fold and <0.5-fold) genes, respectively, in the Δ*sipA* strain. The dotted line indicates the threshold *q* value (0.05). Source data are provided as a Source Data file. **b** Sequence logo of the σ^SsdA-recognizing promoter. The panel is based on 17 promoter sequences among upstream regions of 213 genes upregulated in the Δ*sipA* strain (Fig. S9). The −10 and −35 elements are indicated by dotted rectangles. **c** Purification of recombinant His-σ^SsdA protein. His-σ^SsdA protein was produced in *E. coli*, and the purified protein was analyzed by SDS-PAGE. The separating gel was stained with Coomassie Brilliant Blue (CBB). Molecular size standards are shown in the Marker (M) lane. **d** In vitro transcription assays using the His-σ^SsdA protein. DNA templates covering the promoter regions of *AMIS_25220* (template 1) and *AMIS_68780* (template 2) were prepared using PCR, and in vitro transcription assays were performed using the RNA polymerase core complex from *E. coli* and recombinant His-σ^SsdA protein. The presence (+) and absence (−) of the protein or protein complex are indicated above the panels. Transcripts from the transcriptional start site and terminus of the template are indicated by closed and open triangles, respectively. Schematic diagrams of the template locations are also shown on the left side of the panels. In (**c**) and (**d**), data are representative of similar results obtained in two independent experiments.

S-21) carry a single-nucleotide variant within *ssdA*, although the mutation points are different from strain to strain (Table S2). Considering the phenotypic changes observed in the Δ*sipA*Δ*ssdA* strain, it is most likely that such mutations within *ssdA* restored sporangium dehiscence in these strains via the loss of σ^SsdA function. Thus, we focused on the remaining strain, S-7, because it was expected to carry a mutation(s) in some key gene(s) for sporangium dehiscence other than *ssdA*. Among the seven genes in which mutations were generated in their coding sequences, we focused on *AMIS_37680* because the transcript level of this gene was upregulated 2.4-fold in the Δ*sipA* strain compared to the Δ*sipA*Δ*ssdA* strain (Supplementary Data 2). We designated AMIS_37680 as RsdK (repressor of sporangium dehiscence). A protein database search using the Conserved Domain Database revealed that the 706 amino acid product possesses two GAF_2 domains (pfam13185; residues 48–161 and 326–462), a sensor PAS_4 domain (cl37777; residues 182–299), and a signal transduction histidine kinase domain (COG0642; residues 481–704) (Fig. S10a). Mutant S-7 has a six-amino acid replacement in the PAS_4 domain: $_{273}$TARPII$_{278}$ is replaced with $_{273}$AVRSRP$_{278}$ (Fig. S10a). We assumed that this sequence replacement would impair the function of RsdK, and this assumption is consistent with the results of the gene disruption experiments described below. A gene located just upstream of *rsdK*, *AMIS_37670* (named *rsdR*), encodes a protein of 151 amino acids that possesses a signal transduction response regulator receiver domain (cl19078; residues 16–125) (Fig. S10a). The transcript level of *rsdR* was 2.1-fold higher in the Δ*sipA* strain than in the Δ*sipA*Δ*ssdA* strain (Supplementary Data 2). Although a 97-bp intervening region exists between *rsdR* and *rsdK*, both genes appear to be co-transcribed because their expression was commonly upregulated in the Δ*sipA* strain. Similar transcriptional profiles of *rsdR* and *rsdK* in the transcriptome data also indicated their co-transcription; both the genes are transcribed in the late stage of sporangium formation and during sporangium dehiscence (Fig. S10b). These results strongly indicated that σ^SsdA upregulates the transcription of the *rsdR-rsdK* operon, which encodes a probable two-component regulatory system. However, the *rsdR-rsdK* operon does not seem to be a direct target of σ^SsdA, based on the following observations: the transcript levels of *rsdK* and *rsdR* were upregulated only 2.4- and 2.1-fold, respectively, in the Δ*sipA* strain compared to the Δ*sipA*Δ*ssdA* strain (Supplementary Data 2), and no σ^SsdA-recognizing promoter was found in the upstream region of this operon.

To examine whether the RsdK-RsdR two-component regulatory system controls sporangium dehiscence, we generated a double mutant of *rsdK* and *rsdR* (Δ*rsdK*Δ*rsdR*) by deleting both the genes in the wild-type strain. The Δ*rsdK*Δ*rsdR* strain was not different from wild-type strain by macroscopic observation of mycelia or sporangia formed on YBNM and HAT agar (data not shown). We observed mycelia and sporangia of the mutant grown on HAT agar using SEM, which produced normal sporangia (Fig. S3e). We then induced sporangium dehiscence by suspending and incubating the sporangia harvested from HAT agar in 25 mM histidine solution and observed them using phase-contrast microscopy. Surprisingly, the sporangia of the

Δ*rsdK*Δ*rsdR* strain became transparent and released zoospores much earlier than those of wild-type strain (Fig. 5a–g; Fig. S11a–g). To support this observation, we quantified spores released from sporangia of the wild-type and Δ*rsdK*Δ*rsdR* strains, both of which contained pTYM19-Apra, by incubating a portion of the zoospores on YBNM agar at 30°C and counting the resulting colonies. Considering the phenotypic changes observed by phase-contrast microscopy, we collected zoospores 20 and 60 min after induction of sporangium dehiscence. When we collected zoospores 20 min after the induction of sporangium dehiscence, the number of colonies of the Δ*rsdK*Δ*rsdR* strain was over 10 times higher than that of the wild-type strain (Fig. 5s). Colonies with almost the same numbers were formed in the wild-type and Δ*rsdK*Δ*rsdR* strains when zoospores were collected 60 min after induction (Fig. 5s). In a gene complementation test, normal sporangium dehiscence was restored by introducing pTYM19-Apra containing both *rsdK* and *rsdR* into the Δ*rsdK*Δ*rsdR* strain (Fig. 5h–k, t; Fig. S11h–k).

Next, we generated a double mutant of *sipA* and *rsdR* (Δ*sipA*Δ*rsdR*) by deleting most of the coding sequence of *rsdR* from Δ*sipA* strain. As described above, sporangium dehiscence was severely inhibited in the Δ*sipA* strain (Fig. 1d–f). In contrast, by suspending and incubating the mutant sporangia in 25 mM histidine solution, sporangia of the Δ*sipA*Δ*rsdR* strain opened normally and released spores (Fig. 5l–n; Fig. S11l–n). These results demonstrate that indirect transcriptional activation of *rsdK* and *rsdR* by σ^SsdA is responsible for deficiency in sporangium dehiscence in the Δ*sipA* strain.

Finally, to overexpress these genes, we introduced an integration plasmid carrying the *rsdR-rsdK* operon along with its native promoter into the wild-type strain. Under dehiscence-inducing conditions, 1 h after suspension, most sporangia of the wild-type strain harbouring this *rsdRK*-expressing plasmid did not open to release spores, (Fig. 5o–r; Fig. S11o–r). In this strain, no change in sporangia was observed, even after overnight incubation under dehiscence-inducing conditions (data not shown). Thus, we quantified the zoospores released from sporangia by collecting them 1 h after the induction of sporangium dehiscence. Consistent with microscopic observations, the number of colonies of the wild-type strain harbouring the *rsdRK*-expressing plasmid was reduced to less than one-tenth of that of the wild-type strain harbouring the empty vector (Fig. 5u). These results support the notion that the RsdK-RsdR two-component regulatory system functions as a repressor of sporangium dehiscence.

### σ^SsdA also activates genes required for oxidative stress response

We found a cluster of seven genes, *AMIS_47320* to *AMIS_47380*, among the genes upregulated in Δ*sipA* strain (Supplementary Data 2), whose gene products were predicted to mitigate oxidative stress response because *AMIS_47330* and *AMIS_47380* encode a putative manganese-containing catalase and glutamate-cysteine ligase, respectively. The former catalyses decomposition of hydrogen peroxide to water and oxygen[44], whereas the latter is the rate-limiting enzyme for glutathione biosynthesis[45]. In addition, two σ^SsdA-dependent genes encode a putative catalase (AMIS_37880) and glutamate-cysteine ligase (AMIS_39380), as

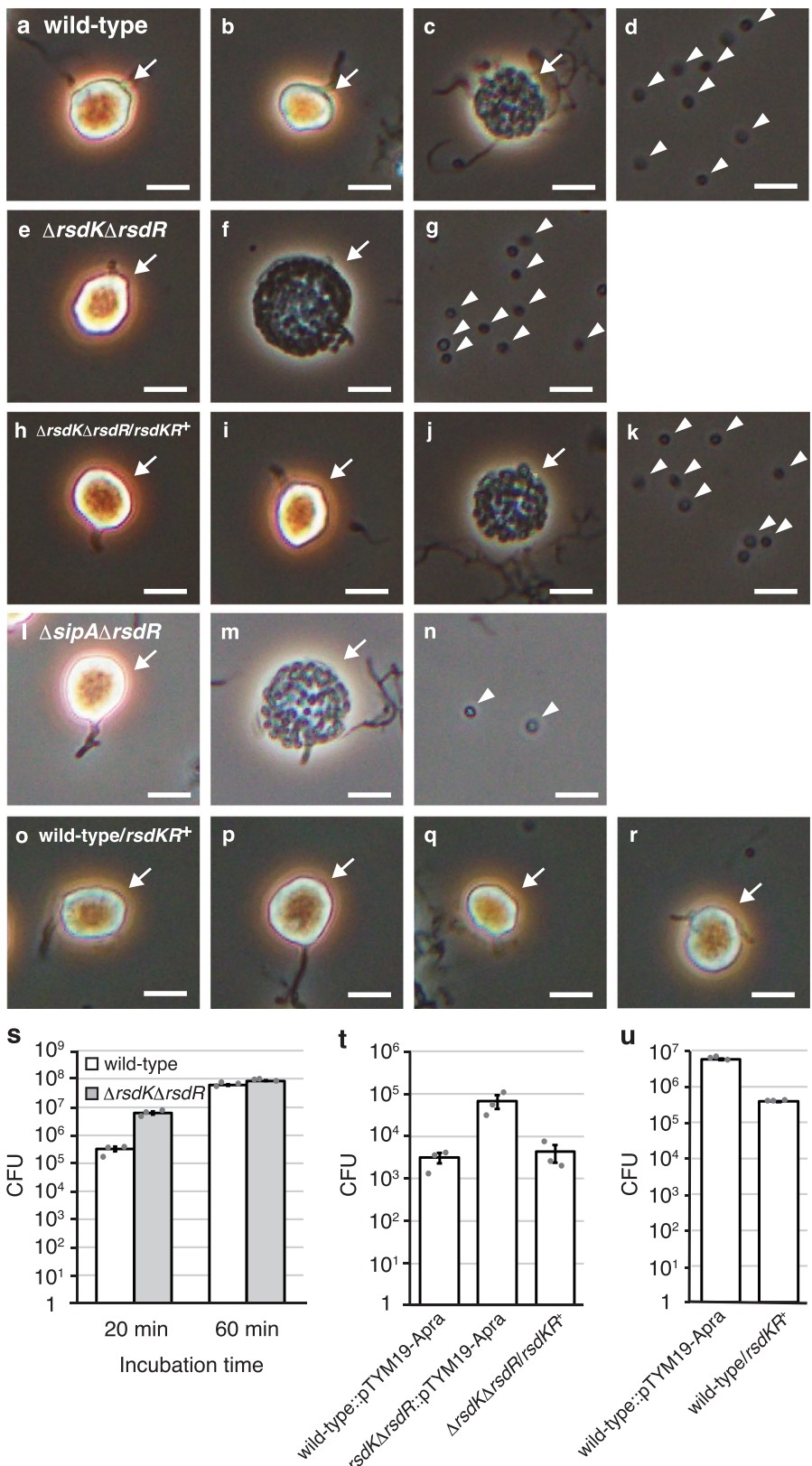

described above. A putative glutamate-cysteine ligase (AMIS_72560) is also encoded by a gene upregulated in the Δ*sipA* strain (Supplementary Data 2). These data suggest that σ$^{SsdA}$ confers tolerance to oxidative stress in the sporangiospores. Therefore, we examined the resistance of the wild-type and Δ*ssdA* strains, both of which contained pTYM19-Apra, to oxidative stress by incubating zoospores in the absence or presence

of 0.03% hydrogen peroxide for 1 h, followed by cultivation on YBNM agar. Wild-type zoospores released from a single HAT agar plate formed over $10^6$ and $10^5$ colonies in the absence and presence of hydrogen peroxide, respectively. However, although the Δ*ssdA* zoospores formed a similar number of colonies (>$10^6$) to the wild-type zoospores in the absence of hydrogen peroxide, they formed only approximately $10^2$

**Fig. 5 | Involvement of the RsdK-RsdR two-component regulatory system in sporangium dehiscence. a–r** Observation of sporangia and zoospores using phase-contrast microscopy. Sporangia produced on HAT agar were harvested and suspended in 25 mM histidine solution to induce sporangium dehiscence. Microscopic images of the wild-type strain (**a–d**), Δ*rsdK*Δ*rsdR* strain (**e–g**), Δ*rsdK*Δ*rsdR* strain harbouring the complementation plasmid (**h–k**), Δ*sipA*Δ*rsdR* strain (**l–n**), and wild-type strain harbouring *rsdR-rsdK*-expressing plasmid (**o–r**). Images in panels **a**, **e**, **h**, **l**, **o** were obtained immediately after suspension. Images in panels **b**, **f**, **i**, **m**, **p** were obtained 5 min after suspension. Images in panels **c**, **g**, **j**, **n**, **q** were obtained 15 min after suspension. Images in panels **d**, **k**, **r** were obtained 30 min after suspension. Sporangia (including those whose membranes became transparent) and released spores are indicated by arrows and arrowheads, respectively. Scale bars, 5 μm. The entire images of each microscopic field are shown in Fig. S11.

**s–u** Number of spores released from the sporangia. Each strain was cultivated on HAT agar at 30 °C for 7 days, and zoospores released from the sporangia formed on one HAT agar plate by pouring 25 mM $NH_4HCO_3$ solution were counted as CFU on YBNM agar. Data are means of three biological replicates ± standard error. Source data are provided as a Source Data file. **s** Number of spores released from sporangia of wild-type and Δ*rsdK*Δ*rsdR* strains. Zoospores were collected 20 and 60 min after pouring $NH_4HCO_3$ solution. **t** Number of spores released from sporangia of wild-type and Δ*rsdK*Δ*rsdR* strains, both of which contained pTYM19-Apra, and Δ*rsdK*Δ*rsdR* strain harbouring the complementation plasmid. Zoospores were collected 20 min after pouring $NH_4HCO_3$ solution. **u** Number of spores released from the wild-type strain harbouring pTYM19-Apra or *rsdR-rsdK*-expressing plasmid. Zoospores were collected 60 min after pouring $NH_4HCO_3$ solution.

colonies in the presence of hydrogen peroxide, clearly indicating that the Δ*ssdA* zoospores were more sensitive to oxidative stress than those of the wild-type strain (Fig. 6). In a gene complementation test, zoospores of the Δ*ssdA* strain harbouring an *ssdA*-expressing plasmid formed a similar number of colonies as the wild-type strain even in the presence of hydrogen peroxide (Fig. 6). These results demonstrate that σ^SsdA is responsible for the oxidative stress response of sporangiospores, probably via direct or indirect transcriptional activation of *AMIS_37880*, *AMIS_39380*, *AMIS_72560*, and the *AMIS_47320–47380* cluster. Notably, glutathione, but not mycothiol, seems to function as a reducing agent for the oxidative stress response in *A. missouriensis* (see Supplementary Note 1). We assume that σ^SsdA is unlikely to be involved in the oxidative stress response during vegetative growth because the transcript levels of *ssdA* and σ^SsdA-dependent genes are very low during vegetative growth (Fig. S8). Thus, these results support our assumption that the function of σ^SsdA is analogous to that of the stationary phase-specific sigma factor σ^S.

## Discussion

In this study, using classical forward genetic methods, we identified the SipA-σ^SsdA sigma/anti-sigma system that is involved in oxidative stress responses of sporangiospores in *A. missouriensis*. We also identified the two-component regulatory system RsdK-RsdR as a key factor in repressing initiation of sporangium dehiscence. Although further studies are required to elucidate detailed molecular mechanisms underlying these regulatory factors, the present study serves as a milestone in the regulation of sporangiospore dormancy and awakening in *A. missouriensis*. In particular, we believe this study is significant in that it clarified the following two points: (I) the presence of a sigma factor that is presumably involved in physiological maturation (not morphological maturation) of sporangium and sporangiospore, including acquisition of oxidative stress resistance, and (II) the presence of a molecular mechanism that delays sporangium dehiscence. The predicted gene regulatory system that is presumably involved in physiological maturation in sporangiospores is schematically represented in Fig. 7.

With regard to (I), it is obvious that σ^SsdA confers oxidative stress tolerance on sporangiospores because mutant zoospores of the Δ*ssdA* strain showed much higher sensitivity to hydrogen peroxide. In contrast, since no other phenotypic changes were observed between the wild-type and Δ*ssdA* strains, the exact nature of the other functions of σ^SsdA is not clear. However, the Δ*sipA* strain, in which σ^SsdA exerts its functions more prominently than in the wild-type strain, showed a sporangium dehiscence-deficient phenotype. Because disruption of *rsdR* in the Δ*sipA* strain resulted in complete restoration of sporangium dehiscence, deficiency in sporangium dehiscence in mutant Δ*sipA* can be attributed to enhanced expression of the *rsdR-rsdK* operon (the *rsdR-rsdK* operon is approximately upregulated 2-fold in the Δ*sipA* strain). Although σ^SsdA probably indirectly enhances transcription of the *rsdR-rsdK* operon, this result indicates that σ^SsdA may induce resistance to sporangium dehiscence. However, we cannot exclude the possibility that σ^SsdA itself is not related to sporangium dehiscence; in

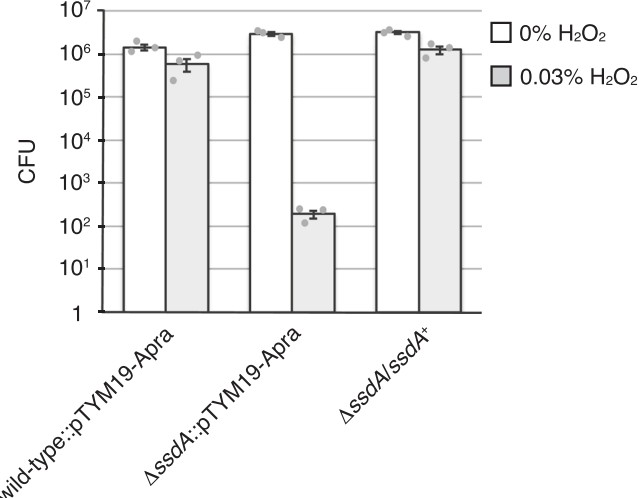

**Fig. 6 | Oxidative stress resistance of zoospores.** Zoospores released from sporangia formed on one HAT agar plate of wild-type and Δ*ssdA* strains, both of which contained pTYM19-Apra, and the Δ*ssdA* strain harbouring the complementation plasmid were incubated in the absence or presence of 0.03% hydrogen peroxide for 1 h and cultivated on YBNM agar at 30 °C for 2 days. The number of colonies was counted, and mean values ± standard error from three biological replicates are shown. Source data are provided as a Source Data file.

other words, the possibility that the sporangium dehiscence-deficient phenotype observed in the Δ*sipA* strain was due to sigma factor competition, in which an increased or decreased amount of a sigma factor leads to lower or higher expression of genes controlled by other sigma factors[46,47]. σ^SsdA is predicted to be in an unrestricted state in the Δ*sipA* strain and this "hyper-active" σ^SsdA may decrease the expression of genes whose transcription is dependent on other sigma factors via sigma factor competition. Nevertheless, we believe that σ^SsdA is involved in physiological maturation of sporangium and sporangiospore including oxidative stress responses, as described in the following paragraph.

Transcription of *ssdA* and *sipA*, as well as 17 direct target genes of σ^SsdA, occurs not only during sporangium dehiscence, but also during and/or after sporangium formation. In addition, many genes were differentially expressed between Δ*sipA* and Δ*sipA*Δ*ssdA* strains on day 6 of incubation on HAT agar. These results clearly indicate that σ^SsdA functions in sporangiospores during and after sporangium formation. Therefore, we assumed that the function of σ^SsdA is analogous to that of stationary phase-specific sigma factor σ^S in *E. coli*[40,41]. σ^S activates 30 or more genes expressed during transition to the stationary phase and in response to various stressors. Target genes of σ^S are involved in various cellular functions, such as protection against DNA damage, morphological changes, osmoprotection, and thermotolerance, which are

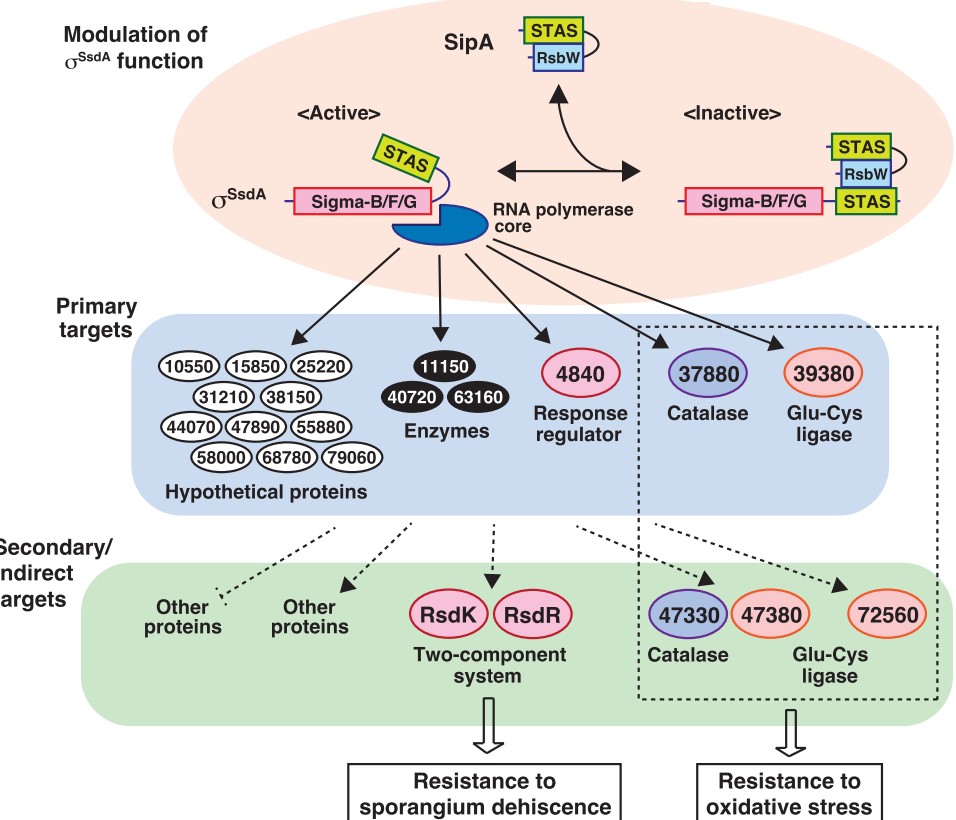

**Fig. 7 | Proposed regulatory model of gene expression by the SipA-σ^SsdA sigma/anti-sigma system.** The anti-sigma (RsbW-like) and anti-sigma factor antagonist (STAS) domains of SipA constitutively interact with each other. To inactivate σ^SsdA, the RsbW-like domain of SipA binds to the STAS domain of σ^SsdA, thereby modulating the expression of the genes under the control of σ^SsdA. Arrows indicate positive control and a line with a vertical short line indicates negative control. Indirect regulation was indicated by dotted lines. Open arrows indicate involvement of gene products in biological phenomena described in the boxes. The prefix "AMIS" is omitted from the locus tag (gene number)-derived protein names.

useful for long-term survival in the stationary phase. We speculated that σ^SsdA also activates the genes involved in sporangiospore survival in the sporangium. However, as described above, we were not able to observe any phenotypic changes, other than oxidative stress response, between wild-type and ΔssdA strains. In this regard, we believe that the laboratory environment is so stable that sporangiospores are well-maintained even in the absence of activation of genes involved in maintenance of dormancy in the ΔssdA strain. Importantly, the assumption that the function of σ^SsdA is analogous to that of σ^S involves a major conceptual shift with respect to sporangiospores of *A. missouriensis*: sporangiospores may not be completely dormant cells and several genes should be transcribed for the maintenance of their "apparently dormant" state. Although further analysis is required to confirm this assumption, it should be noted that *A. missouriensis* sporangiospores differ from exospores of the genus *Streptomyces* as well as endospores of *B. subtilis*. Most importantly, sporangiospores are protected by the sporangium matrix and sporangium membrane; therefore, they seem to retain a considerable amount of water, in contrast to *B. subtilis* spores, in which the water content drops below 35% of the wet weight[4,48,49]. Thus, a low level of gene expression can be maintained for a relatively long period in the sporangiospores. Considering the actual growth environment, it is unlikely that dormancy continues beyond one year because sporangium dehiscence is probably triggered by water during rainfall. In terms of cell dormancy, the sporangium of *A. missouriensis* may be more similar to the hibernation state of an animal rather than a plant seed.

Notably, with regard to (II), sporangium dehiscence does not occur synchronously. The timing of onset of sporangium dehiscence varied from sporangium to sporangium when sporangia scraped from HAT agar were suspended in histidine solution and observed under a phase-contrast microscope. Based on this phenomenon, we assume that each sporangium has a different ability to resist sporangium dehiscence under dehiscence-inducing conditions. This may be of ecological importance because sporangia that do not open easily might be more advantageous for reproduction than having all sporangia initiating sporangium dehiscence at the same time. Interestingly, no substances other than water seem to be required for sporangium dehiscence (i.e., revival of sporangiospores) in *A. missouriensis* in contrast to *B. subtilis*, where substances known as germinants are necessary for spore awakening (see Supplementary Note 2). In our study, we demonstrated that the two-component regulatory system RsdK-RsdR induces resistance to sporangium dehiscence; introduction of only one additional copy of the *rsdR-rsdK* operon into the wild-type strain resulted in complete absence of sporangium dehiscence. Furthermore, in the Δ*rsdK*Δ*rsdR* strain, sporangium dehiscence occurred immediately after incubation of sporangia in 25 mM histidine solution, which was much earlier than that in the wild-type strain. Because RsdR has only a signal transduction response regulator receiver domain, it may exert its function (i.e., repression of the initiation of sporangium dehiscence) via protein-protein interactions with its target protein, probably after being phosphorylated by transmission of phosphate from self-phosphorylated RsdK. Identification of the target of RsdR is the next challenge. Environmental conditions that promote self-phosphorylation of RsdK should also be examined. We also need to confirm whether the RsdK-RsdR regulatory system functions more efficiently during/after sporangium formation on HAT agar or during sporangium dehiscence under dehiscence-inducing conditions.

Notably, both SipA and $\sigma^{SsdA}$ harbour an anti-sigma factor antagonist domain in their N- and C-terminal portions, respectively. Both anti-sigma factor antagonist domains bind to the C-terminal anti-sigma factor domain of SipA. Initially, we assumed that these two anti-sigma factor antagonist domains compete for binding to the anti-sigma factor domain of SipA and that the $\sigma^{SsdA}$/SipA pair is controlled via an unusual switching mechanism of domain-domain interactions. However, based on the structural model (Fig. 3, Fig. S7), this seems unlikely; the anti-sigma factor domain of SipA binds to each of the two anti-sigma factor antagonist domains on opposite faces (Fig. 3). Some post-translational modifications of SipA or other molecules (proteins, ions, compounds, etc.) may be involved in the formation or dissociation of the SipA−$\sigma^{SsdA}$ complex. To the best of our knowledge, no proteins with domain structures similar to those of SipA or $\sigma^{SsdA}$ have been previously reported. Particularly, the domain structure of $\sigma^{SsdA}$ is interesting because it is the first sigma factor that has an anti-sigma factor antagonist domain and because the cognate anti-sigma factor SipA seems to inhibit the function of $\sigma^{SsdA}$ by binding to the anti-sigma factor antagonist domain but not to the sigma factor domain. In addition, our experiments indicated that an amino acid replacement in the anti-sigma factor antagonist domain (L372P) resulted in the loss of the sigma factor function of $\sigma^{SsdA}$. Therefore, the mechanism of $\sigma^{SsdA}$ inhibition by SipA seems to be unique. The molecular mechanism underlying functional regulation of $\sigma^{SsdA}$ by SipA is also a future challenge.

Orthologues of *sipA* and *ssdA* are present in genome sequences of the genera *Actinoplanes*, *Pseudosporangium*, *Couchioplanes*, and *Micromonospora*, which belong to the family *Micromonosporaceae* (Fig. S12). In species of the genera *Actinoplanes*, *Pseudosporangium*, and *Couchioplanes*, *sipA* homologues are located on the gene locus adjacent to *ssdA* homologues (Fig. S12). In contrast, a gene encoding cytochrome c oxidase subunit I (*ctaD*) is located between *sipA* and *ssdA* homologues in species of the genus *Micromonospora* (Fig. S12). Beyond the family *Micromonosporaceae*, both *sipA* and *ssdA* homologues have been found in *Nonomuraea* sp. TT08I-71, which belongs to the family *Streptosporangiaceae*. In this strain, a *ctaD* homologue is also located between the *sipA* and *ssdA* homologues (Fig. S12). Because *sipA* and *ssdA* homologues are not found in other actinomycetes, we postulate that the regulatory system involving SipA and $\sigma^{SsdA}$ homologues has evolved among sporangium-forming actinomycetes, mostly in the family *Micromonosporaceae*.

In conclusion, this study provides insights into spore biology. We propose that $\sigma^{SsdA}$ activates its regulon involved in physiological maturation of sporangium and sporangiospore including oxidative stress responses. Among the genes upregulated in the presence of "hyper-active" $\sigma^{SsdA}$, we identified *rsdK* and *rsdR*, which encode a two-component regulatory system that plays a pivotal role in the repression of sporangium dehiscence. In our laboratory, a more detailed analysis of the RsdK-RsdR two-component regulatory system is in progress to obtain a more complete picture of the downstream steps. Thus, this study also provides important clues to the regulatory mechanisms involved in the revival of inert bacterial sporangiospores.

## Methods
### General methods
Bacterial strains, plasmid vectors, and media used in this study have been described previously[26,28,50]. Primers used in this study are listed in Table S3. *A. missouriensis* cells were prepared as described previously[35]. SEM was performed using an S-4800 scanning electron microscope (Hitachi, Tokyo, Japan) as described previously[51]. Phase-contrast microscopy observations of sporangia and zoospores were performed using a BH-2 light microscope (Olympus, Tokyo, Japan) as previously described[34]. Free zoospores were quantified as described previously[52].

### Sporangiospore and zoospore heat resistance assays
Wild-type sporangia formed on one HAT agar plate were harvested using a spatula and suspended in 1 ml of distilled water. The suspension was then divided into two equal parts. A sporangium dehiscence-inducing solution (500 µl), which was prepared by pouring 25 mM $NH_4HCO_3$ solution (10 ml) onto an aseptic HAT agar plate (9 cm in diameter) and collecting the solution after incubation for 1 h, was added to one part, while the same amount of distilled water was added to the other part. After rotation for 1 h at room temperature, we confirmed that almost all sporangia suspended in the dehiscence-inducing solution opened and released spores, using phase-contrast microscopy. Both suspensions were then incubated at 50 °C in a water bath for 90 min. Every 30 min, a portion of the suspension was retrieved and transferred into ice-cold water (200 µl) in a new tube. The samples were inoculated onto YBNM agar and incubated at 30 °C for 2 days. A portion of the suspension before incubation at 50 °C was inoculated onto YBNM agar in parallel. From the number and size of colonies formed on YBNM agar, the respective numbers of both sporangia and released spores that survived heat stress were estimated because a colony grown from a sporangium was much larger than that grown from a released spore.

### Isolation of sporangium dehiscence-deficient strains
Wild-type mycelia were cultivated on HAT agar at 30 °C for 7 days for sporangium formation. Then, 25 mM $NH_4HCO_3$ solution (10 ml/plate) was poured onto the HAT agar plate, followed by incubation for 1 h to induce sporangium dehiscence. The solution was collected from the surface of the agar plate and filtered through a 5-µm membrane filter (Pall Corporation, NY, USA) to eliminate mycelia and sporangia. The resulting zoospore-containing solution was irradiated with UV light until the survival rate of the zoospores reached approximately 1%. Using the irradiated solution, a series of steps for sporangium formation, dehiscence, enrichment of the dehiscence-deficient sporangia, and inoculation onto HAT agar were conducted as follows: (i) the zoospore (or heat-treated sporangium)-containing solution was inoculated onto HAT agar and cultivated at 30 °C for 7 days for sporangium formation; (ii) sporangia were harvested from the agar surface with a spatula and suspended in the dehiscence-inducing solution, followed by rotation at room temperature for 90 min to induce sporangium dehiscence; (iii) the suspension was incubated in a water bath at 50 °C for 30 min, followed by cooling in ice-cold water; and (iv) the suspension was diluted appropriately and inoculated onto HAT agar. After repeating steps (i) to (iv) twice or more, a portion of the solution after step (iii) was inoculated on YBNM agar and cultivated at 30 °C for 2 days. Single colonies were picked and streaked on YBNM agar and the plate was incubated at 30 °C for 2 days. Each isolated strain was inoculated into peptone-yeast extract-magnesium (PYM) broth and cultivated at 30 °C for 2 days. After washing with 0.75% (w/v) NaCl solution, the mycelia were inoculated on HAT agar and cultivated at 30 °C for 7 days for sporangium formation. Sporangium dehiscence of each strain was analyzed using phase-contrast microscopy, and sporangium dehiscence-deficient strains were isolated.

### Isolation of suppressor strains
The parental strain M-12 was inoculated and cultivated on HAT agar at 30 °C for 7 days for sporangium formation. Next, 25 mM $NH_4HCO_3$ solution was poured onto the HAT agar plate to induce sporangium dehiscence. After incubation for 1 h, the solution was collected from the surface of the agar plate and filtered through a 5-µm membrane filter (Pall Corporation) to eliminate mycelia and sporangia. The resulting zoospore-containing solution was irradiated with UV light until the survival rate of the zoospores reached approximately 7%. Using the irradiated solution, sporangium formation, dehiscence, and enrichment of the released zoospores were conducted as follows: (i) the zoospore-containing solution was inoculated on HAT agar and

cultivated at 30 °C for 7 days to induce sporangium formation; (ii) the sporangia were harvested from the agar surface with a spatula and suspended in 25 mM histidine solution, followed by rotation at room temperature for 1 h to induce sporangium dehiscence; and (iii) the suspension was filtered through a 5-μm membrane filter to eliminate mycelia and sporangia. After repeating steps (i) to (iii) twice or more, a portion of the solution was inoculated onto YBNM agar and cultivated at 30 °C for 2 days. Then, single colonies were picked up to streak on YBNM agar and incubated at 30 °C for 2 days. Each of the isolated strains was inoculated into PYM broth and cultivated at 30 °C for 2 days. After washing with 0.75% NaCl solution, mycelia were inoculated on HAT agar and cultivated at 30 °C for 7 days for sporangium formation. The sporangium dehiscence of each strain was analyzed by suspending and incubating the sporangia harvested from the surface of HAT agar in 25 mM histidine solution. Strains whose sporangia opened and released spores under dehiscence-inducing conditions were isolated.

## Genome sequencing of isolated strains

The strains isolated by either of the two enrichment strategies were inoculated into PYM broth and cultivated at 30 °C for 2 days. Genomic DNA was extracted using the CTAB method[25]. Sequencing libraries were prepared using 3 μg of DNA as the starting material, and sequencing was performed using a HiSeq 2500 sequencer (Illumina, CA, USA). Library construction and sequencing were performed by Novogene (Beijing, China). Sequencing reads were filtered by sequence quality and mapped to the *A. missouriensis* genome sequence using CLC Genomics Workbench (Illumina).

## Construction of gene deletion mutants

To construct gene deletion mutants, the upstream and downstream regions of the target gene were amplified using PCR. The amplified DNA fragments were digested with *Eco*RI and *Xba*I (for upstream regions) or *Xba*I and *Hin*dIII (for downstream regions), and cloned into pUC19 digested with the same restriction enzymes. The generated plasmids were sequenced to confirm that no PCR-derived errors were present. The cloned fragments were digested with *Eco*RI and *Xba*I (for upstream regions) and *Xba*I and *Hin*dIII (for downstream regions) and cloned together into pK19mob*sacB*[53], whose kanamycin resistance gene had been replaced with the apramycin resistance gene *aac(3)IV*[28], digested with *Eco*RI and *Hin*dIII. The generated plasmids were introduced into *A. missouriensis* by conjugation as described previously[29]. Apramycin-resistant colonies resulting from single crossover recombination were isolated. One of them was cultivated in PYM broth at 30 °C for 48 h, and the mycelia suspended in 0.75% NaCl solution were spread onto Czapek-Dox broth agar medium (BD, NJ, USA) containing extra sucrose (final concentration 5%). After incubation at 30 °C for 5 days, sucrose-resistant colonies were inoculated onto YBNM agar with or without apramycin to confirm that they were sensitive to apramycin. Apramycin-sensitive and sucrose-resistant colonies resulting from the second crossover recombination were isolated as candidates for the gene deletion mutant. Disruption of the target genes was confirmed by PCR (data not shown).

## Construction of strains for gene complementation tests

The 1.0-, 1.5-, and 3.0-kbp DNA fragments containing the promoter and coding sequences of *sipA*, *ssdA*, and *rsdK-rsdR*, respectively, were amplified by PCR. The amplified fragments were digested with *Eco*RI and *Hin*dIII and cloned into pTYM19-Apra[35,54] digested with the same restriction enzymes. The generated plasmids were sequenced to confirm that no PCR-derived errors were present, and were introduced into the Δ*sipA*, Δ*ssdA*, Δ*sipA*Δ*ssdA*, or Δ*rsdK*Δ*rsdR* mutants by conjugation, as described previously[28]. Plasmid pTYM19-Apra was also introduced into the wild-type and mutant strains for the vector control strains. Apramycin-resistant colonies were obtained.

## BACTH assay

The bacterial adenylate cyclase-based two-hybrid assay was conducted using the BACTH system kit (Euromedex, Strasbourg, France) according to the manufacturer's instructions. To construct the T18 or T25 domain fusion plasmids, the coding sequences of *sipA* and *ssdA* were amplified using PCR. The DNA fragments were digested with *Bam*HI and *Pst*I, and cloned into pUC19 digested with the same restriction enzymes. The generated plasmids were sequenced to confirm that no PCR-derived errors were introduced. The cloned fragments were digested with *Bam*HI and *Pst*I and cloned into the vectors pKT25 (for *sipA*) and pUT18C (for *sipA* or *ssdA*) digested with the same restriction enzymes. *E. coli* BTH101 cells were co-transformed with the T18 and T25 domain fusion plasmids, and transformants were selected on LB agar containing ampicillin and kanamycin. At least three individual colonies per assay were grown overnight at 30 °C in LB broth containing ampicillin and kanamycin. The cultures were inoculated into LB broth containing ampicillin, kanamycin, and isopropyl β-D-1-thiogalactopyranoside (IPTG) and cultivated at 30 °C for 48 h. β-Galactosidase activity was quantified as previously described[34].

## AlphaFold- and AlphaFold-Multimer-based predictions of protein structure

The SipA and SipA-σ^SsdA heterodimer complex structures were predicted by AlphaFold2 and AlphaFold-Multimer, respectively, using ColabFold (v1.5.3) (https://colab.research.google.com/github/sokrypton/ColabFold/blob/main/AlphaFold2.ipynb) with default parameters[55]. For the predictions using the STAS and RsbW-like domains of SipA, the amino acid sequences from the N-terminal end to Glu-124 and from Val-118 to the C-terminal end, respectively, were used. For predictions using the STAS domain of σ^SsdA, the amino acid sequence from Ala-258 to the C-terminal end was used. The generated structures (Supplementary Data 1) were evaluated using the predicted local distance difference test (pLDDT) score (for full-length SipA), and the predicted aligned error (PAE) score (for heterodimer complexes composed of SipA and σ^SsdA) and the most accurate structure for each was visualized and coloured using PyMol (Schrödinger, NY, USA).

## RNA-Seq and in silico analysis

RNAs were extracted from the wild-type, Δ*sipA*, and Δ*sipA*Δ*ssdA* strains, as described previously[35]. The quality and quantity of total RNAs were assessed using the Bioanalyzer DNA1000 (Agilent Technologies, CA, USA). Sequencing libraries were prepared with 3 μg of RNA as the starting material, and sequencing was performed using the MiSeq or HiSeq 2500 sequencer (Illumina, CA, USA) to generate non-directional single-read 50-nucleotide (wild-type strain) or paired-end 150-nucleotide reads (Δ*sipA* and Δ*sipA*Δ*ssdA* strains). Library construction and sequencing of the Δ*sipA* and Δ*sipA*Δ*ssdA* strains were performed by Novogene (Beijing, China). The reads were filtered by sequence quality and mapped to the *A. missouriensis* genome sequence using CLC Genomics Workbench (Illumina).

## Preparation of His-σ^SsdA

A 1.1-kbp DNA fragment containing the *ssdA* coding sequence was amplified by PCR. The fragment was digested with *Eco*RI and *Hin*dIII, and cloned into pUC19 digested with the same restriction enzymes to generate pUC19-*ssdA*. pUC19-*ssdA* was sequenced to confirm the absence of PCR-derived errors. The cloned fragment was digested with *Nde*I and *Hin*dIII, and cloned into pColdII (Takara Biochemicals, Shiga, Japan) digested with the same restriction enzymes to generate pColdII-*ssdA*. pColdII-*ssdA* was introduced into *E. coli* BL21(DE3) cells. The transformants were cultivated in LB broth (100 ml) at 37 °C for 2.5 h and at 15 °C for 30 min. Then, IPTG was added to the culture to a final concentration of 0.1 mM. After further cultivation at 15 °C for 30 h, the cells were collected by centrifugation at 3000 × g for 10 min, suspended in 5 ml of lysis buffer (50 mM NaH₂PO₄, 300 mM NaCl, 10 mM

imidazole, 10% glycerol, pH 8.0), and disrupted by sonication. After centrifugation at $10,000 \times g$ for 30 min, His-$\sigma^{SsdA}$ was purified from the supernatant using Ni-nitrilotriacetic acid Superflow resin (Qiagen, Tokyo, Japan), according to the manufacturer's instructions. His-$\sigma^{SsdA}$ was eluted with elution buffer (50 mM $NaH_2PO_4$, 300 mM NaCl, 500 mM imidazole, 10% glycerol, pH 8.0). The quality of the purified protein was assessed by sodium dodecyl sulphate-polyacrylamide gel electrophoresis (SDS-PAGE).

### In vitro transcription assay

In vitro transcription run-off assays were performed using $[\alpha-^{32}P]$-CTP (30 TBq/mmol) purchased from PerkinElmer (Waltham, MA, USA) as described previously[43]. DNA fragments containing the promoter and coding sequences of *AMIS_25220* (0.5 kb) or *AMIS_68780* (0.4 kb) were amplified using PCR. The fragments were digested with *Eco*RI and *Hin*dIII, and cloned into pUC19 digested with the same restriction enzymes. The generated plasmids were sequenced to confirm that no PCR-derived errors were present. The cloned fragments were digested with *Eco*RI and *Hin*dIII, and used as templates. Approximately 30 pmol of His-$\sigma^{SsdA}$ and 1 unit of *E. coli* RNA polymerase core enzyme (New England Biolabs, MA, USA) were used for the reactions.

### Oxidative stress assay

Wild-type and mutant strains were cultivated on HAT agar at 30 °C for 7 days for sporangium formation. Then, 25 mM $NH_4HCO_3$ solution was poured onto HAT agar and incubated at room temperature for 1 h to induce sporangium dehiscence. The zoospore-containing solution was collected from the surface of HAT agar and filtered through a 5-μm membrane filter to eliminate mycelia and sporangia. Then, 30% hydrogen peroxide solution was added to the zoospore-containing solution at a final concentration of 0.03% and incubated at room temperature for 1 h. For control experiments, distilled water was added instead of the hydrogen peroxide solution. The zoospores were collected by centrifugation and suspended in 0.75% (w/v) NaCl solution to eliminate hydrogen peroxide. A portion of the suspension was inoculated onto YBNM agar and cultivated at 30 °C for 2 days. The number of colonies formed on YBNM agar was counted.

### Reporting summary

Further information on research design is available in the Nature Portfolio Reporting Summary linked to this article.

## Data availability

The Conserved Domain Database (https://www.ncbi.nlm.nih.gov/Structure/cdd/cdd.shtml) was used for the search of protein domains. The MEME (http://meme-suite.org/tools/meme) and FIMO (http://meme-suite.org/tools/fimo) algorithms were used for the search of conserved sequence motifs. Nucleotide sequence data of the RNA-Seq analysis have been deposited in the DDBJ Sequence Read Archive under the accession numbers DRA012687 and DRA016947. PDB files for predicted structures are provided in Supplementary Data 1. Other data supporting the findings of this study are available within the article and its supplementary material. Source data are provided with this paper.

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

## Acknowledgements

This research was supported in part by Grants-in-Aid for Scientific Research (A) (JP26252010; to Y.O.), (B) (JP18H02122; to Y.O.), and (C) (JP17K07711 and JP20K05781; to T.T.), and a Grant-in-Aid for Scientific Research on Innovative Areas (JP19H05685; to Y.O.) from the Ministry of Education, Culture, Sports, Science, and Technology of Japan. This work was also supported in part by the Japan Society for the Promotion of Science (JSPS) A3 Foresight Program (to Y.O.). Exhaustive transcriptome analysis was supported by JSPS KAKENHI Grant Number JP16H06279 (PAGS; to Y.O.).

## Author contributions

T.T., K.M., and R.D. performed experiments. All authors designed the study and analyzed the data. T.T. and Y.O. wrote the manuscript. Y.O. supervised this study.

## Competing interests

The authors declare no conflicts of interest.
