## [Peer Review File · Nature Communications]

A unique sigma/anti-sigma system in the actinomycete
Actinoplanes missouriensisReviewer #1 (Remarks to the Author):

This is a very well-written and clearly presented piece of work that is noteworthy and original for several reasons. First, it describes the discovery of a bacterial sigma factor/anti-sigma factor pair with unique and broadly interesting features, where both the anti-sigma and the sigma itself contain STAS domains, which normally are found in anti-anti-sigma factors. The data suggest that the STAS domains in this case mediate the interactions between the anti-sigma SipA and the sigma SsdA and are instrumental in the regulation of the system. A self-interaction of the internal STAS domain with the RsbW-like anti-sigma domain may keep the SipA inactive, while switching to an interaction between RsbW-like domain of SipA with the STAS on SsdA (instead of the normally observed binding to the sigma factor domain in these type of proteins) is suggested to mediate the inactivation of sigma SsdA.

Further, the study identifies new regulators of the intriguing developmental life cycle of the actinomycete *Actinoplanes missouriensis*. Thus, new information about an original but broadly interesting bacterial developmental system. Specifically, the study concerns the development of sporangiospores, release of sporangiospores from sporangia and conversion into motile zoospores, and physiological features of these spores, including a demonstration that sigma SsdA mediates the development of resistance to oxidative stress. Finally, it is shown that a two-component system (RsdK/RsdR), which is indirectly regulated at transcription level by SipA/SsdA, in some way downregulates spore dehiscence (i.e. opening of sporangia to release the hundred or more spores).

The work is logical and clearly described, including clear descriptions of methods, data, and conclusions, and well supported by supplementary data. It uses nice genetic strategies (enrichment and isolation of mutants and suppressor mutants) as well as transcriptomics to identify several previously unknown regulators and genes involved in the release and features of zoospores. The conclusions are generally well supported by the data, but a few aspects should be considered and revised

Comments:

1. Could parts of the phenotype and the extensive effects on gene expression patterns in sipA mutants be due to "sigma factor competition", i.e. too high activity of SsdA may lead to lower or altered expression of genes controlled by other sigma factors that are important in later stages of development in this bacterium? The observation that sigma SsdA is required for the phenotype of the sipA mutant (e.g. page 10) would be consistent with a model where unrestricted activity of SsdA in the sipA mutant strains could lead to indirect effects (via sigma factor competition for the core RNA polymerase) on expression of genes that depend on other sigma factors. This possible explanation should be taken into account and discussed. See for example Nyström (2004) *Mol. Microbiol.* 54: 855–862 and Nandy (2022) *Microbiology* 168: 001195.

2. The proposal that sigma SsdA is involved regulating maintenance of "dormant state of sporangiospores, both directly and indirectly" (e.g. lines 530-531) should be reconsidered. The data support that SsdA is involved in oxidative stress resistance of the spores, but it is not clear whether SsdA has any direct role in maintenance of dormancy or awakening from dormancy. The ssdA mutant itself shows no clear phenotype, and the only reported defect of the mutant is the sensitivity to oxidative stress.

3. The interactions between the domains of SipA and SsdA are only shown here by BACTH assays. Although the data are clear and consistent, the method has its limitations. It would therefore have strengthened the paper if the interactions between the relevant domains could have been demonstrated and verified by other methods. If not by testing direct interactions with purified proteins, there is now also the possibility to probe interactions with for example AlphaFold Multimer predictions.

Various minor comments:

4. The interpretation of the test of heat resistance of sporangiospores and zoospores (Fig. S1) may be misleading. What appears to be compared are sporangia, containing 100-200 spores each, versus free zoospores, and results are evaluated with determination of cfu. The killing kinetics will

be very different when comparing “clumps” of large numbers of cells versus single cells. It is therefore highly questionable whether the conclusion that sporangiospores are more heat-resistant than zoospores (lines 115-117) is correct. The experiment and the figure are still valuable for the paper since they pave the ground for the mutant enrichment strategy, but the conclusion in lines 115-117 should be reconsidered and rephrased.

5. It is not correct to say that sporangia reflect light in phase-contrast microscopy. They are likely appearing very bright in phase contrast because they have a high refractive index, leading to reversal of contrast (similarly to what is seen with for example mature endospores) in phase-contrast microscopy. This should be adjusted at several places in text in figure legends, e.g. lines 81-82.

6. The series of representative microscopy image in Figs. 1 and 4 are very clear. But given that it also is mentioned that sporangium dehiscence does not occur synchronously. (line 562), I wonder if there is any statistics here. Is it possible to say, for example, how large fraction of detected sporangia are phase-bright (like Fig. 1a) and how many have lost this feature and look like in Fig. 1b at the tested time points? Similarly, does Fig. 4 o-r show complete abolishment of sporangium dehiscence, or was there some fraction of the sporangia that opened up also at these tested times?

7. Lines 169-170: It says sipA mutant sporangia released 100-fold lower numbers of spores than WT. But are all the counted colonies coming from released spores? Would not intact sporangia that have not released any spores also give colonies. It would help the reader if this was clarified.

8. In Fig. 5 and line 486-489, it would be useful to mention or show how many cfu were obtained when spores were not subjected to oxidative stress.

9. There has been some progress regarding germination receptors and the awakening from dormancy in *Bacillus* endospores that may be relevant to cite in the Introduction, e.g. Gao et al. (2023) *Science* 380, 387–391.

10. Regarding hsdR-hsdK it says that sigma SsdA activates expression of these genes, although indirectly. Just to avoid giving false impression of a mechanism of regulation, perhaps change “activate” to “up-regulate” in this context, e.g. lines 424 and 457.

11. New data are presented in the Discussion when Fig. S8, referred to in line 544. It is up to the journal policy whether this is allowed, but it would easily be solved by describing also these results in the Results sections.

12. Small language thing in line 242: Change “than” to “compared to” to make the sentence correct.

13. In legends to figures with quantifications of viable spores, it should be made clear the numbers shown are CFU resulting from sporangia per HAT agar plate.

14. Fig. S6 is referred to before Fig. S5

Reviewer #2 (Remarks to the Author):

The work of Tezuka and colleagues is carefully done, and describes the role of two different regulatory systems in controlling the release of zoospores from the sporangium of *Acinoplanes missouriensis*. The first system comprised a sigma factor-anti-sigma factor pair (SsdA-SipA), where each protein contained multiple domains, and interactions were mediated by both SsdA sigma and SipA anti-sigma domains, as well as anti-sigma-antagonist domains (associated with both proteins) and the anti-sigma domains of SipA, although what controlled the different domain

interactions is not clear. The transcriptional effects stemming from altered SsdA activity were extensive, and a subset of affected genes were determined to be possible direct targets. Phenotypically, loss of the anti-sigma factor SipA led to premature release of spores from the sporangium (presumably due to hyperactivity of the SsdA sigma factor), while loss of *ssdA* led to reduced resistance to oxidative stress. Included within the broad (indirect) regulon of SsdA was a two-component regulatory system termed RsdR/RsdK, and this system also appeared to function in repressing spore release from the sporangium.

The discovery of these two new regulatory systems governing spore release in the Actinoplanes is interesting. The manuscript would be further strengthened by additional biochemical investigations probing the regulatory interplay between the SsdA-SipA domains, and/or establishing a mechanistic basis for the control of spore release via RsdR.

1) Line 178: While Ser 229 may well be essential for SidA, this seems to be a slight overinterpretation of the data presented to this point. A Pro-Ser substitution at this position is not functional, so what this suggests is that a Pro at this position leads to a protein that is not functional – but other residues may well function at this position. It would be worth softening this conclusion to accommodate these other possibilities.

Lines 245-248 and 263-265 and 279-281: While the point mutant variant does indeed appear to fail to interact with the sigma factor, it could be because the resulting protein is unstable – in which case it is not interacting, as it is not present at sufficient levels. Again, it would be worth tempering the language/interpretation of the effect of this mutation, in the absence of something like immunoblots to demonstrate protein levels in the mutant relative to wild type.

2) More of a comment than anything else: the suppressor mutation-based search for target sigma factors of SidA ended up being a successful approach; however, it isn't clear why the authors didn't start out by looking at the predicted functions/domains of adjacent genes, as sigma factor-anti-sigma factor pairs are often encoded together in the chromosome (and *ssdA* – whose product is the target of SipA – ended up being found right upstream).

3) Line 253-255: Given the ease of protein prediction using AlphaFold these days, is it possible to predict what the effect of this mutation would be on the protein structure? Do the data indeed support a loss of function for this particular mutation? – does the structure change dramatically or are there significant differences predicted for interaction with its different partner domains?

4) The interaction between the SipA STAT domain and the anti-sigma factor domain seems to be much less strong than the anti-sigma factor domain and the SsdA STAT domain. Could the authors develop a model or propose testable hypotheses around the significance of this and how it could impact the regulation of SsdA activity?

5) Why not do ChIP-seq to identify direct targets for sigma SsdA?

6) The spacing of the proposed -10 and -35 sequences recognized by sigma SsdA is unusual, as the two sequences would be found on opposite sides of the DNA helix without the additional activity of some binding protein that promotes significant supercoiling or unwinding. Additional commentary about this would be helpful.

7) I don't fully understand the two different products indicated by arrowheads in Fig. 3D. Also, was a negative control experiment done for the in vitro transcription experiments, using a promoter region that was not expected to be targeted by SsdA?

8) The profiles of the proposed SsdA target genes in Figure S5 share both similarities and significant differences. For example, the two genes selected for in vitro transcription show similar profiles during sporangium formation, but show opposing profiles during dehiscence, where 25220 levels dropped, and the levels of 68670 rose significantly. Do the authors have an explanation for this?

Editorial comments:

1) Line 309: ssgA or ssdA?

2) Line 313-314: Not clear why details on the RNA-seq analyses will be published elsewhere? It would be worth removing this statement, as it seems that sufficient detail to understand the experiment as presented here.

Reviewer #3 (Remarks to the Author):

The authors are investigating a very interesting morphological differentiation process and have developed a useful screen to identify genes involved in sporangium dehiscence. This led to the identification of a sigma factor/ anti-sigma factor couple (SsdA/SipA) that is proposed to play a role in this process, along with the further identification of a two-component system (RsdR/RsdK). The experiments are well designed and presented and there are some interesting aspects of the system, not least the very unusual structure of the sigma factor that suggests some complex signalling interplay. Further the RsdR-RsdK couple look interesting and appear to have a negative role in dehiscence, which unfortunately is not further expanded.

However, I have a major concern that the central claim of the paper, that the SsdA/SipA couple play a role in dehiscence is flawed and based on a misinterpretation of the data. The authors have shown convincingly that both point and null mutations SipA prevents normal dehiscence. This suggests a role but is not conclusive since mutation of a negative regulator (if SipA is an anti-sigma factor) can result in overexpression of the negatively regulated gene, which can lead to pleiotropic/indirect effects. In the case of sigma factor/anti-sigma factor couples, it is not uncommon that the mutation of an anti-sigma factor can lead to the hyperactivity of the sigma factor, which can outcompete other sigma factors leading to a phenotype unrelated to the gene couple. For example, it is possible that the phenotype is caused by the outcompeting of a different sigma factor involved in differentiation. An example of this can be seen in the actinobacterium *Streptomyces coelicolor* where mutation of the anti-sigma involved in controlling the oxidative stress sigma factor SigR results in a block in sporulation at a very specific point – however SigR is not involved in sporulation (doi: 10.1046/j.1365-2958.2001.02298.x).

This possibility is strengthened by the authors ability to isolate suppressors in the cognate sigma factor SsdA and appears to be confirmed by the finding that null mutations in this sigma factor do not have a dehiscence related phenotype. As stated by the authors "These observations indicated that the deletion of ssdA had no effect on sporangium formation and dehiscence under the tested conditions." Therefore, while much of the work is solid, the conclusion that SsdA/SipA is involved in dehiscence is likely flawed.

Another minor point is that the authors should show biochemically that SipA has anti-sigma activity – ie it inhibits SsdA in in vitro transcription assays.

Point-by-point response to the reviewers' comments

Reviewer #1

This is a very well-written and clearly presented piece of work that is noteworthy and original for several reasons. First, it describes the discovery of a bacterial sigma factor/anti-sigma factor pair with unique and broadly interesting features, where both the anti-sigma and the sigma itself contain STAS domains, which normally are found in anti-anti-sigma factors. The data suggest that the STAS domains in this case mediate the interactions between the anti-sigma SipA and the sigma SsdA and are instrumental in the regulation of the system. A self-interaction of the internal STAS domain with the RsbW-like anti-sigma domain may keep the SipA inactive, while switching to an interaction between RsbW-like domain of SipA with the STAS on SsdA (instead of the normally observed binding to the sigma factor domain in these type of proteins) is suggested to mediate the inactivation of sigma SsdA.

Further, the study identifies new regulators of the intriguing developmental life cycle of the actinomycete *Actinoplanes missouriensis*. Thus, new information about an original but broadly interesting bacterial developmental system. Specifically, the study concerns the development of sporangiospores, release of sporangiospores from sporangia and conversion into motile zoospores, and physiological features of these spores, including a demonstration that sigma SsdA mediates the development of resistance to oxidative stress. Finally, it is shown that a two-component system (RsdK/RsdR), which is indirectly regulated at transcription level by SipA/SsdA, in some way downregulates spore dehiscence (i.e. opening of sporangia to release the hundred or more spores).

The work is logical and clearly described, including clear descriptions of methods, data, and conclusions, and well supported by supplementary data. It uses nice genetic strategies (enrichment and isolation of mutants and suppressor mutants) as well as transcriptomics to identify several previously unknown regulators and genes involved in the release and features of zoospores. The conclusions are generally well supported by the data, but a few aspects should be considered and revised.

* Thank you very much for your careful reading and positive comments.

Comments:

1. Could parts of the phenotype and the extensive effects on gene expression patterns in sipA mutants be due to “sigma factor competition”, i.e. too high activity of SsdA may

lead to lower or altered expression of genes controlled by other sigma factors that are important in later stages of development in this bacterium? The observation that sigma SsdA is required for the phenotype of the sipA mutant (e.g. page 10) would be consistent with a model where unrestricted activity of SsdA in the sipA mutant strains could lead to indirect effects (via sigma factor competition for the core RNA polymerase) on expression of genes that depend on other sigma factors. This possible explanation should be taken into account and discussed. See for example Nyström (2004) Mol. Microbiol. 54: 855–862 and Nandy (2022) Microbiology 168: 001195.

* Thank you very much for this significant comment. We did not fully consider sigma factor competition as a possible cause for the phenotypic changes observed in the $\Delta sipA$ strain. We agree with this reviewer that the sporangium dehiscence-deficient phenotype observed in the $\Delta sipA$ strain may be due to sigma factor competition induced by the unrestricted activity of σ^{SsdA} . Therefore, we have added a description of sigma factor competition in the Discussion section of the revised manuscript as follows (L. 589-597). Please also see our response to the comment from the Reviewer #3.

“However, we cannot exclude the possibility that σ^{SsdA} itself is not related to sporangium dehiscence; in other words, the possibility that the sporangium dehiscence-deficient phenotype observed in the $\Delta sipA$ strain was due to sigma factor competition, in which an increased or decreased amount of a sigma factor leads to lower or higher expression of genes controlled by other sigma factors (46, 47). σ^{SsdA} is predicted to be in an unrestricted state in the $\Delta sipA$ strain and this “hyper-active” σ^{SsdA} may decrease the expression of genes whose transcription is dependent on other sigma factors via sigma factor competition.”

2. The proposal that sigma SsdA is involved regulating maintenance of “dormant state of sporangiospores, both directly and indirectly” (e.g. lines 530-531) should be reconsidered. The data support that SsdA is involved in oxidative stress resistance of the spores, but it is not clear whether SsdA has any direct role in maintenance of dormancy or awakening from dormancy. The *ssdA* mutant itself shows no clear phenotype, and the only reported defect of the mutant is the sensitivity to oxidative stress.

* Thank you very much for this constructive criticism. We agree with this reviewer that it is not clear whether σ^{SsdA} directly regulates the maintenance of dormancy in sporangiospores because the sole phenotypic change observed in the $\Delta ssdA$ strain was

higher sensitivity to oxidative stress. In our previous manuscript, we noted this fact repeatedly as described below.

“In contrast, since no other phenotypic changes were observed between the wild-type and $\Delta ssdA$ strains, the exact nature of the other functions of σ^{SsdA} is not clear.”

“We speculated that σ^{SsdA} also activates the genes involved in sporangiospore survival in the sporangium. However, as described above, we were not able to observe any phenotypic changes, other than oxidative stress response, between wild-type and $\Delta ssdA$ strains.”

However, we had assumed that σ^{SsdA} should be involved in the suppression of sporangium dehiscence based on the experimental results that sporangium dehiscence did not occur at all in the $\Delta sipA$ strain. If sigma factor competition can explain this, it is possible that σ^{SsdA} is not involved in suppressing sporangium dehiscence.

According to the comments from the reviewer and Reviewer #3, we concluded that we must avoid insisting on the relationship between σ^{SsdA} and spore dormancy (or repression of sporangium dehiscence). Therefore, we have changed the manuscript title and revised many sentences concerning this point (see revised manuscript). We have also revised Fig. 6 (Proposed regulatory model of gene expression by the SipA- σ^{SsdA} sigma/anti-sigma system) according to this decision.

With these points carefully considered, we would like to insist on the involvement of σ^{SsdA} in the physiology of sporangiospores including oxidative stress response, and would like to propose its possible involvement in dormancy maintenance as one of its roles (see revised manuscript).

Please also see our response to Reviewer #3.

3. The interactions between the domains of SipA and SsdA are only shown here by BACTH assays. Although the data are clear and consistent, the method has its limitations. It would therefore have strengthened the paper if the interactions between the relevant domains could have been demonstrated and verified by other methods. If not by testing direct interactions with purified proteins, there is now also the possibility to probe interactions with for example AlphaFold Multimer predictions.

* Thank you very much for this constructive suggestion. We tried but unfortunately failed to produce a recombinant SipA protein in its soluble form using *E. coli* as a host.

Thus, we could not investigate the direct interaction between SipA and σ^{SsdA} by using purified proteins. However, according to this comment, we examined the interaction using the AlphaFold-Multimer. As a result, the interaction between the anti-sigma factor (RsbW-like) domain of SipA and the anti-anti-sigma factor (STAS) domain of σ^{SsdA} was reliably predicted. Furthermore, the AlphaFold-Multimer predicted that the anti-sigma factor domain of SipA binds to each of the two anti-sigma factor antagonist domains of SipA and σ^{SsdA} on opposite faces. These predictions suggested that the anti-sigma factor domain of SipA interacts with the two anti-sigma factor antagonist domains in different manners and that the anti-sigma factor antagonist domains of SipA and σ^{SsdA} do not compete for binding to the anti-sigma factor domain of SipA. We have added the description of this prediction in the Results and Discussion sections and Fig. S7 in the revised manuscript (see revised manuscript).

Various minor comments:

4. The interpretation of the test of heat resistance of sporangiospores and zoospores (Fig. S1) may be misleading. What appears to be compared are sporangia, containing 100-200 spores each, versus free zoospores, and results are evaluated with determination of cfu. The killing kinetics will be very different when comparing “clumps” of large numbers of cells versus single cells. It is therefore highly questionable whether the conclusion that sporangiospores are more heat-resistant than zoospores (lines 115-117) is correct. The experiment and the figure are still valuable for the paper since they pave the ground for the mutant enrichment strategy, but the conclusion in lines 115-117 should be reconsidered and rephrased.

* Thank you very much for this important comment. We agree with this reviewer that we had better not to conclude that sporangiospores are more resistant to heat stress than zoospores from the experimental result shown in Fig. S1. We have added the description of this experimental result in the Results section as follows (L. 110-116).

“As expected, the survival rates of sporangiospores (spores in a sporangium), determined every 30 min after heat treatment, were much higher than those of zoospores (spores released from sporangia) (Fig. S1). Based on this apparent heat tolerance of sporangiospores, we developed a scheme for mutant screening in which sporangium dehiscence-deficient strains were enriched from a mutant library generated via UV irradiation of wild-type zoospores.”

In response to this comment, we would like to describe one aspect of this experiment. As described in the legend of Fig. S1, a large portion of sporangiospores in the sporangium seemed to be alive when the sporangium survived the heat treatment. (A sporangium contains 100-200 spores, and therefore forms a much larger colony than a zoospore. Only large colonies were observed even after the heat treatment.) We speculate that if the sporangia are damaged enough to kill the spores inside by heat treatment, it is not a situation in which some spores survive but in which all spores die.

5. It is not correct to say that sporangia reflect light in phase-contrast microscopy. They are likely appearing very bright in phase contrast because they have a high refractive index, leading to reversal of contrast (similarly to what is seen with for example mature endospores) in phase-contrast microscopy. This should be adjusted at several places in text in figure legends, e.g. lines 81-82.

* Thank you very much for this important information. We misunderstood the phase-contrast microscopic images of the sporangia. We have changed the description of the microscopic images in the revised manuscript in the Introduction and the legend of Fig. 1 as follows (L. 77-80; L. 1088-1089).

“Under the latter conditions, the sporangia appear phase-bright immediately after suspension and then the sporangium membrane gradually becomes transparent before spore release when observed by phase-contrast microscopy (see Fig. 1a–c).”

“Immediately after suspension, the sporangia appeared phase-bright (a, d, g, j, m, p, and s).”

6. The series of representative microscopy image in Figs. 1 and 4 are very clear. But given that it also is mentioned that sporangium dehiscence does not occur synchronously (line 562), I wonder if there is any statistics here. Is it possible to say, for example, how large fraction of detected sporangia are phase-bright (like Fig. 1a) and how many have lost this feature and look like in Fig. 1b at the tested time points? Similarly, does Fig. 4o-r show complete abolishment of sporangium dehiscence, or was there some fraction of the sporangia that opened up also at these tested times?

* Thank you very much for this constructive suggestion. Although the process of sporangium dehiscence does not completely proceed synchronously as mentioned in the

manuscript, most sporangia show similar morphological changes under the conditions used in this experiment. We have added the entire images of the microscopic fields in Figs. S4 and S11 in the revised manuscript. As shown in Fig. S11, most sporangia of the wild-type strain harbouring an additional copy of the *rsdR-rsdK* operon did not open and remained phase-bright during the time course of this experiment (Fig. S11o-r). We have added the information in the legends of Figs. 1 and 4 as follows (L. 1092; L. 1146-1147).

“The entire images of each microscopic field are shown in Fig. S4.”

“The entire images of each microscopic field are shown in Fig. S11.”

7. Lines 169-170: It says sipA mutant sporangia released 100-fold lower numbers of spores than WT. But are all the counted colonies coming from released spores? Would not intact sporangia that have not released any spores also give colonies. It would help the reader if this was clarified.

* In this experiment, we poured 10 ml of 25 mM NH_4HCO_3 solution to one sporangium-forming HAT agar plate and incubated the plate for 1 h to induce sporangium dehiscence. After collected zoospore-containing solution from the plate, we filtrated the solution through a 5- μm membrane filter to eliminate hyphae and sporangia. Through the procedure, no hyphae or sporangia were observed in the filtrate when observed by phase-contrast microscopy. Therefore, we consider that all colonies formed on the YBNM agar plate came from the zoospores released from sporangia in both the wild-type and ΔsipA strains. We have added the information to the Results section as follows (L. 165-168).

“Because the solution containing the zoospores released from sporangia was filtered through a 5- μm membrane filter to eliminate hyphae and sporangia in this experiment (see Methods), all colonies formed on YBNM agar came from the spores released from sporangia (35).”

8. In Fig. 5 and line 486-489, it would be useful to mention or show how many cfu were obtained when spores were not subjected to oxidative stress.

* We have added the experimental results obtained in the absence of hydrogen peroxide to the Results section and Fig. 5 as follows (L. 543-554).

“Therefore, we examined the resistance of the wild-type and $\Delta ssdA$ strains, both of which contained pTYM19-AprA, to oxidative stress by incubating zoospores in the absence or presence of 0.03% hydrogen peroxide for 1 h, followed by cultivation on YBNM agar. Wild-type zoospores released from a single HAT agar plate formed over 10^6 and 10^5 colonies in the absence and presence of hydrogen peroxide, respectively. However, although the $\Delta ssdA$ zoospores formed a similar number of colonies ($>10^6$) to the wild-type zoospores in the absence of hydrogen peroxide, they formed only approximately 10^2 colonies in the presence of hydrogen peroxide, clearly indicating that the $\Delta ssdA$ zoospores were more sensitive to oxidative stress than those of the wild-type strain (Fig. 5). In a gene complementation test, zoospores of the $\Delta ssdA$ strain harbouring an *ssdA*-expressing plasmid formed a similar number of colonies as the wild-type strain even in the presence of hydrogen peroxide (Fig. 5).”

9. There has been some progress regarding germination receptors and the awakening from dormancy in *Bacillus* endospores that may be relevant to cite in the Introduction, e.g. Gao *et al.* (2023) *Science* 380, 387–391.

* We have cited the paper in the Introduction as suggested.

10. Regarding hsdR-hsdK it says that sigma SsdA activates expression of these genes, although indirectly. Just to avoid giving false impression of a mechanism of regulation, perhaps change “activate” to “up-regulate” in this context, e.g. lines 424 and 457.

* We have changed the word “activate” into “upregulate” as suggested.

11. New data are presented in the Discussion when Fig. S8, referred to in line 544. It is up to the journal policy whether this is allowed, but it would easily be solved by describing also these results in the Results sections.

* We have transferred the description of the results shown in Fig. S5 to the Results section as suggested (L. 216-222).

12. Small language thing in line 242: Change "than" to "compared to" to make the sentence correct.

* Thank you very much for your careful reading. We have changed the word as suggested.

13. In legends to figures with quantifications of viable spores, it should be made clear the numbers shown are CFU resulting from sporangia per HAT agar plate.

* We have added the information to the legends of the Figs. 1, 4, 5, S2, and S5 as suggested.

14. Fig. S6 is referred to before Fig. S5

* Thank you very much for your careful reading, but we had already referred to the figures in the correct order (Figs. S8 and S9 in the revised manuscript).

Reviewer #2

The work of Tezuka and colleagues is carefully done, and describes the role of two different regulatory systems in controlling the release of zoospores from the sporangium of *Acinoplanes missouriensis*. The first system comprised a sigma factor-anti-sigma factor pair (SsdA-SipA), where each protein contained multiple domains, and interactions were mediated by both SsdA sigma and SipA anti-sigma domains, as well as anti-sigma-antagonist domains (associated with both proteins) and the anti-sigma domains of SipA, although what controlled the different domain interactions is not clear. The transcriptional effects stemming from altered SsdA activity were extensive, and a subset of affected genes were determined to be possible direct targets.

Phenotypically, loss of the anti-sigma factor SipA led to premature release of spores from the sporangium (presumably due to hyperactivity of the SsdA sigma factor), while loss of *ssdA* led to reduced resistance to oxidative stress. Included within the broad (indirect) regulon of SsdA was a two-component regulatory system termed RsdR/RsdK, and this system also appeared to function in repressing spore release from the sporangium.

The discovery of these two new regulatory systems governing spore release in the Actinoplanes is interesting. The manuscript would be further strengthened by additional biochemical investigations probing the regulatory interplay between the SsdA-SipA domains, and/or establishing a mechanistic basis for the control of spore release via RsdR.

* Thank you very much for your positive comments. We consider that the regulatory mechanism controlling the interaction between SipA and σ^{SsdA} and the identification of a target factor(s) of RsdR are important subjects of our future research. We tried but unfortunately failed to produce a recombinant SipA protein in its soluble form using *E. coli* as a host, which hampered biochemical investigations in this stage.

1) Line 178: While Ser 229 may well be essential for SidA, this seems to be a slight overinterpretation of the data presented to this point. A Pro-Ser substitution at this position is not functional, so what this suggests is that a Pro at this position leads to a protein that is not functional – but other residues may well function at this position. It would be worth softening this conclusion to accommodate these other possibilities.

* Thank you very much for your careful reading. We agree with this reviewer that the substitution of Ser-229 with amino acids other than Pro may have no effect on the SipA function. We have changed the sentence in the revised manuscript as follows (L. 176-179).

“Sporangium dehiscence and the number of spores released from the sporangia were not restored by the introduction of this mutated gene (Fig. 1j–l, v; Fig. S4j–l), indicating that the S229P mutation renders SipA non-functional with respect to these phenotypic changes.”

Lines 245-248 and 263-265 and 279-281: While the point mutant variant does indeed appear to fail to interact with the sigma factor, it could be because the resulting protein is unstable – in which case it is not interacting, as it is not present at sufficient levels. Again, it would be worth tempering the language/interpretation of the effect of this mutation, in the absence of something like immunoblots to demonstrate protein levels in the mutant relative to wild type.

* We agree with this reviewer that there is a possibility that the SipA (S229P) variant is unstable. In this respect, we predicted the structure of a complex composed of SipA and σ^{SsdA} by using AlphaFold-Multimer (please see our response to the comment 3 from the Reviewer #1). According to the predicted structures, the anti-sigma factor domain of SipA binds to the two anti-sigma factor antagonist domains of SipA and σ^{SsdA} on opposite faces, and Ser-229 of SipA is located at the interface between the anti-sigma factor and anti-sigma factor antagonist domains of SipA (Fig. S7 in the revised manuscript). Therefore, it is difficult to describe the reason why the S229P mutation repressed the interaction between the anti-sigma factor domain of SipA and the anti-sigma factor antagonist domain of σ^{SsdA} based on the predicted structures. Thus, we speculate that the S229P variant is less stable than the wild-type SipA, abolishing the complex formation between SipA and σ^{SsdA} . We have added the description of these results in the revised manuscript as follows (L. 253-262, see revised manuscript for the prediction of the structure of the SipA- σ^{SsdA} complex).

“We did not detect any interactions between SipA (S229P) and σ^{SsdA} (Fig. 2a), supporting the notion that the SipA (S229P) variant is not functional in *A. missouriensis* because of its inability to repress σ^{SsdA} function. There are two possible explanations for the negative effect of S229P replacement on the function of SipA: (i) Ser-229 plays a significant role in the interaction between SipA and σ^{SsdA} , and the SipA (S229P) variant cannot bind to σ^{SsdA} ; and (ii) S229P replacement considerably decreases the stability of SipA, and the structurally unstable SipA (S229P) variant cannot exert its function. Based on the predicted structure of the SipA- σ^{SsdA} complex, we believe that the latter explanation is more probable (see below).”

2) More of a comment than anything else: the suppressor mutation-based search for target sigma factors of SidA ended up being a successful approach; however, it isn't clear why the authors didn't start out by looking at the predicted functions/domains of adjacent genes, as sigma factor-anti-sigma factor pairs are often encoded together in the chromosome (and *ssdA* – whose product is the target of SipA – ended up being found right upstream).

* Because SipA possesses both anti-sigma factor and anti-sigma factor antagonist domains, we were not confident that SipA functions as an anti-sigma factor when we identified SipA as an important factor for sporangium dehiscence. Thus, we conducted suppressor screening as described in the manuscript. As a consequence, our

experimental strategy strongly suggested that σ^{SsdA} is the sole target of SipA, which was supported by further experimental evidence. Therefore, we believe that our approach is better than focusing directly on the gene loci adjacent to *sipA*.

3) Line 253-255: Given the ease of protein prediction using AlphaFold these days, is it possible to predict what the effect of this mutation would be on the protein structure? Do the data indeed support a loss of function for this particular mutation? – does the structure change dramatically or are there significant differences predicted for interaction with its different partner domains?

* Thank you very much for this constructive suggestion. According to this comment, we predicted the structures of the σ^{SsdA} and σ^{SsdA} (S100P) proteins using AlphaFold2, but almost no change was observed between the predicted structures (data not shown). Because Ser-100 is located in Region 2 of the sigma factor domain, which is involved in the interaction between a sigma factor and -10 element of its target promoters, we considered that the replacement of Ser-100 with Pro in σ^{SsdA} inhibits the recognition of target promoters by σ^{SsdA} . We have added this information to the Results section of the revised manuscript as follows (L. 268-271).

“Ser-100 is located in Region 2 of the sigma factor domain of σ^{SsdA} (Fig. S5b), which is involved in the interaction between the sigma factor and -10 element of its target promoters (38). Thus, we speculate that σ^{SsdA} (S100P) is inactive because it fails to recognize its target promoters.”

4) The interaction between the SipA STAT domain and the anti-sigma factor domain seems to be much less strong than the anti-sigma factor domain and the SsdA STAT domain. Could the authors develop a model or propose testable hypotheses around the significance of this and how it could impact the regulation of SsdA activity?

* Thank you very much for this interesting comment. As described above, the AlphaFold-Multimer predicted that the anti-sigma factor domain of SipA binds to each of the two anti-sigma factor antagonist domains of SipA and σ^{SsdA} (tentatively named STAS-1 and STAS-2) on opposite faces. If this is true, it is not surprising that the binding affinities are different between the anti-sigma factor domain/STAS-1 and the anti-sigma factor domain/STAS-2. As described in the Discussion, we think that the

mechanism of σ^{SsdA} inhibition by SipA seems to be unique. The molecular mechanism underlying functional regulation of σ^{SsdA} by SipA is also a future challenge.

5) Why not do ChIP-seq to identify direct targets for sigma SsdA?

* We agree with this reviewer that the ChIP-Seq experiment provides an important piece of information about the direct targets for σ^{SsdA} , but we have not yet conducted the experiment because many efforts will be required to find appropriate experimental conditions for ChIP-Seq of sporangia. Therefore, we think that it may be our future research subject. We believe that our conclusion is convincingly supported by the current experimental evidence.

6) The spacing of the proposed -10 and -35 sequences recognized by sigma SsdA is unusual, as the two sequences would be found on opposite sides of the DNA helix without the additional activity of some binding protein that promotes significant supercoiling or unwinding. Additional commentary about this would be helpful.

* Thank you very much for this important comment. Because RNA polymerase holoenzymes are tolerant of moderate variations in spacer length, a similar spacer length (14–16 bp) had also been reported for the target promoters of σ^{B} in *Streptomyces coelicolor* A3(2), which plays an important role in the osmoprotection and morphological differentiation (Lee *et al.* 2004. *J Microbiol* **42**:147-151). Therefore, we believe that the spacer length in the target promoters of σ^{SsdA} (14–15 bp) is not necessarily unusual. We have added the description of the spacer length in the revised manuscript as follows (L. 412-415).

“Although the spacer length between the -10 and -35 elements of this motif is shorter than that of the target promoters of the principal sigma factor, a similar spacer length (14–16 bp) has been reported for the target promoters of σ^{B} in *Streptomyces coelicolor* A3(2) (42).”

7) I don't fully understand the two different products indicated by arrowheads in Fig. 3D. Also, was a negative control experiment done for the *in vitro* transcription experiments, using a promoter region that was not expected to be targeted by SsdA?

* While *E. coli* RNA polymerase holoenzyme recognizes specific promoter sequence elements, it also interacts with single-stranded DNA in a nonspecific manner and randomly initiates transcription (Kaguni and Kornberg 1982. *J Biol Chem* **257**:5437-5443). Thus, the signals indicated by the open triangles in Fig. 3d seem to be the transcripts from the termini of the templates because we used the DNA fragments digested with restriction enzymes as the templates as described in the Methods section (L. 858-867). In this *in vitro* transcription assay, we also used four DNA fragments covering the upstream regions from *AMIS_4840*, *AMIS_10550*, *AMIS_11150*, and *AMIS_58000* as templates, but only weak transcripts from the transcriptional start sites were detected for *AMIS_4840* and *AMIS_10550*, and no signals from the transcriptional start points were detected for *AMIS_11150* and *AMIS_58000* (data not shown). Considering these results, transcriptional regulatory mechanisms may differ from gene to gene within the 17 target genes of σ^{SsdA} . We predict that other factors, such as transcriptional activators, may be essential for the transcriptional initiation of *AMIS_11150* and *AMIS_58000*. Therefore, the results shown Fig. 3d, in which the transcripts with expected lengths were detected in the presence of His- σ^{SsdA} , proved that σ^{SsdA} can recognize the sequence elements as promoters. Usually, negative controls using template DNAs without a target promoter have not been used in *in vitro* transcription experiments.

8) The profiles of the proposed SsdA target genes in Figure S5 share both similarities and significant differences. For example, the two genes selected for *in vitro* transcription show similar profiles during sporangium formation, but show opposing profiles during dehiscence, where 25220 levels dropped, and the levels of 68670 rose significantly. Do the authors have an explanation for this?

* As described in Fig. 3 and Fig. S9, we demonstrated that σ^{SsdA} initiates the transcription of *AMIS_25220* and *AMIS_68780*. Meanwhile, we do not exclude the possibility that other regulatory factors, including transcriptional regulators, control the transcript levels of these genes in addition to σ^{SsdA} . Therefore, as a possible scenario, the transcription of *AMIS_68780* may be upregulated by a transcriptional activator during sporangium dehiscence. We have added the description of this possibility in the Results section as follows (L. 439-443).

“Meanwhile, the transcription profiles slightly differed for several genes. For instance, the transcript level of *AMIS_25220* decreased during sporangium dehiscence, whereas

that of *AMIS_68780* increased. Unknown transcriptional regulators may contribute to the differences of the transcript levels of these genes.”

Editorial comments:

1) Line 309: *ssgA* or *ssdA*?

* Thank you very much for your careful reading. We have changed the word “*ssgA*” to “*ssdA*” in the revised manuscript.

2) Line 313-314: Not clear why details on the RNA-seq analyses will be published elsewhere? It would be worth removing this statement, as it seems that sufficient detail to understand the experiment as presented here.

* We plan to publish a paper that describes whole transcriptome changes during morphological development of *A. missouriensis* with hierarchical clustering analysis and promoter analysis. This is why we described “details on the RNA-seq analyses will be published elsewhere”. However, as you pointed, this expression may be misleading. Therefore, we have deleted the sentence in the revised manuscript as suggested.

Reviewer #3

The authors are investigating a very interesting morphological differentiation process and have developed a useful screen to identify genes involved in sporangium dehiscence. This led to the identification of a sigma factor/anti-sigma factor couple (*SsdA/SipA*) that is proposed to play a role in this process, along with the further identification of a two-component system (*RsdR/RsdK*). The experiments are well designed and presented and there are some interesting aspects of the system, not least the very unusual structure of the sigma factor that suggests some complex signalling interplay. Further the *RsdR-RsdK* couple look interesting and appear to have a negative role in dehiscence, which unfortunately is not further expanded.

However, I have a major concern that the central claim of the paper, that the *SsdA/SipA* couple play a role in dehiscence is flawed and based on a misinterpretation of the data. The authors have shown convincingly that both point and null mutations *SipA* prevents normal dehiscence. This suggests a role but is not conclusive since mutation of a negative regulator (if *SipA* is an anti-sigma factor) can result in overexpression of the

negatively regulated gene, which can lead to pleiotropic/indirect effects. In the case of sigma factor/anti-sigma factor couples, it is not uncommon that the mutation of an anti-sigma factor can lead to the hyperactivity of the sigma factor, which can outcompete other sigma factors leading to a phenotype unrelated to the gene couple. For example, it is possible that the phenotype is caused by the outcompeting of a different sigma factor involved in differentiation. An example of this can be seen in the actinobacterium *Streptomyces coelicolor* where mutation of the anti-sigma involved in controlling the oxidative stress sigma factor SigR results in a block in sporulation at a very specific point – however SigR is not involved in sporulation (doi: 10.1046/j.1365-2958.2001.02298.x).

This possibility is strengthened by the authors ability to isolate suppressors in the cognate sigma factor SsdA and appears to be confirmed by the finding that null mutations in this sigma factor do not have a dehiscence related phenotype. As stated by the authors “These observations indicated that the deletion of *ssdA* had no effect on sporangium formation and dehiscence under the tested conditions.” Therefore, while much of the work is solid, the conclusion that SsdA/SipA is involved in dehiscence is likely flawed.

* Thank you very much for your constructive criticism of the SipA/ σ^{SsdA} system, which has not been fully considered in the original manuscript. As the reviewer pointed out, point or null mutations in *sipA* seem to render σ^{SsdA} “hyper-active,” which may lead to indirect effects on genes under the control of other sigma factors through sigma factor competition. We agree with the reviewer that the extensive effects on gene expression profiles detected between the $\Delta sipA$ and $\Delta sipA \Delta ssdA$ strains may be, at least in part, due to sigma factor competition. Meanwhile, we believe that the functional model of the SipA/ σ^{SsdA} system proposed in our study is also possible. In particular, we believe that our experimental data indicate that σ^{SsdA} is involved in sporangiospore physiology including oxidative stress response.

According to the comments from the reviewer and Reviewer #1, we concluded that we must avoid insisting on the relationship between σ^{SsdA} and spore dormancy (or repression of sporangium dehiscence). Therefore, we have changed the manuscript title and revised many sentences concerning this point (see revised manuscript). We have also added the description of sigma factor competition in the Discussion section, in which we have described that we cannot exclude the possibility that σ^{SsdA} itself is not related to sporangium dehiscence (see revised manuscript). In addition, we have revised

Fig. 6 (Proposed regulatory model of gene expression by the SipA- σ^{SsdA} sigma/anti-sigma system) according to this decision. With these points carefully considered, we would like to insist on the involvement of σ^{SsdA} in the physiology of sporangiospores, and propose its possible involvement in dormancy maintenance as one of its roles.

Please also see our response to the comments 1 and 2 from the Reviewer #1.

Another minor point is that the authors should show biochemically that SipA has anti-sigma activity – ie it inhibits SsdA in *in vitro* transcription assays.

* Thank you very much for this important suggestion. We tried but unfortunately failed to produce a recombinant SipA protein in its soluble form using *E. coli* as a host. Therefore, we could not perform an *in vitro* transcription assay with SipA. However, we demonstrated that the direct interaction between SipA and σ^{SsdA} by BACTH assays (Figs. 2 and S6 in the revised manuscript) and the negative effects of SipA on the σ^{SsdA} function by genetic analyses (Fig. 1). We believe that our model is convincingly supported by the current experimental data. Please also see our response to the comment 4 from the Reviewer #2.

Reviewer #1 (Remarks to the Author):

Figure 2B: There seems to be some mistake in annotations below the graph. It is currently written as if there are duplications of the test of SipA_C* versus SsdA_C and also duplication of the double negative control. Please correct.

Reviewer #2 (Remarks to the Author):

The authors have done a great job of addressing the original reviewers' comments. I had only a few minor points for their consideration.

- 1) The predicted structure and possible regulatory interplay between SipA and sigma SsdA are fascinating. I would recommend moving the AlphaFold images from Fig. S7 to the main manuscript (specifically the predicted structures of SipA, SsdA and the complex), but leaving the predicted local distance difference test (pLDDT) (b, e, h, k, and n) and predicted aligned error (PAE) data in the supplementary data.
- 2) Line 360: *ssdA* (not *ssgA*)?
- 3) The inability to purify SipA must have been frustrating. It may be worth trying to co-express SsdA and SipA together in *E. coli* instead of just individually.
- 4) Given the potential for sigma factor competition in the *sipA* mutant strain (due to overactive sigma SsdA), amongst the 333 down-regulated genes identified in the RNA-seq experiment, are there any obvious groups of genes contained within these that might constitute the regulation of another sigma factor that was impacted by this competition (recognizing that the sporulating actinobacteria have many sigma factors and the regulons of many these are not well defined)?

Reviewer #3 (Remarks to the Author):

The authors have carefully considered and acted on the reviewers' comments which has made important improvements to the manuscript. Most significantly, the authors have agreed that the sigma/ anti-sigma couple that was identified in a screen for mutants involved in sporangium dehiscence, might not be involved in this process. Rather sigma competition caused by hyperexpression of the sigma factor could account for the phenotypes.

1. Although the authors agree that the phenotype seen in SsdA mutants might be caused by sigma competition they maintain that the regulatory couple are involved somehow in sporangiospore physiology. This is primarily because the genes are expressed during sporangium formation and dehiscence, although presumably the same could be said for any gene that is expressed during this process. The fact that double mutants are not affected in either process suggests that any physiological role is minimal. Ultimately, even though the title of the paper has changed to reflect the likelihood that the sigma/anti-sigma system is involved in oxidative stress response rather than sporangiogenesis/dehiscence, the paper retains a focus on the latter and therefore is somewhat misleading to a reader looking for insights into this fascinating morphological development process. The study does identify a two-component system that looks likely to be involved in dehiscence, but is not likely to be SsdA/SipA controlled and is only studied briefly.
2. The domain organisation of the sigma and anti-sigma couple are certainly interesting and suggests complex regulatory mechanism, the inability to isolate soluble SipA means that it is uncertain how these domains interact and how a signal, such as oxidative stress, is perceived.
3. The paper now shows that SsdA/SipA is responsible for oxidative stress resistance in zoospores, however it is not mentioned whether the system is important during vegetative growth. If so, this would argue for a more general role of the system in oxidative stress rather than a specific role in sporangiogenesis.

Point-by-point response to the reviewers' comments

Reviewer #1

Figure 2B: There seems to be some mistake in annotations below the graph. It is currently written as if there are duplications of the test of SipA_C* versus SsdA_C and also duplication of the double negative control. Please correct.

* Thank you very much for your careful reading. We have corrected the Fig. 2b.

Reviewer #2

The authors have done a great job of addressing the original reviewers' comments. I had only a few minor points for their consideration.

1) The predicted structure and possible regulatory interplay between SipA and sigma SsdA are fascinating. I would recommend moving the AlphaFold images from Fig. S7 to the main manuscript (specifically the predicted structures of SipA, SsdA and the complex), but leaving the predicted local distance difference test (pLDDT) (b, e, h, k, and n) and predicted aligned error (PAE) data in the supplementary data.

* Thank you very much for this constructive suggestion. We agree with this reviewer that the structures of SipA and σ^{SsdA} predicted by AlphaFold and AlphaFold-Multimer are fascinating because they suggested unprecedented binding manners in the sigma/anti-sigma system. Although we think that further experimental evidence is required for the complex structure, we have presented the predicted structure of SipA/ σ^{ssdA} complex in new Fig. 3, as suggested.

2) Line 360: ssdA (not ssgA)?

* Thank you very much for your careful reading. We have changed the word as suggested (L. 359).

3) The inability to purify SipA must have been frustrating. It may be worth trying to co-express SsdA and SipA together in *E. coli* instead of just individually.

* Thank you very much for this constructive suggestion. We consider that the co-expression of *sipA* and *ssdA* is an important subject of our future research.

4) Given the potential for sigma factor competition in the sipA mutant strain (due to overactive sigma SsdA), amongst the 333 down-regulated genes identified in the RNA-seq experiment, are there any obvious groups of genes contained within these that might constitute the regulon of another sigma factor that was impacted by this competition (recognizing that the sporulating actinobacteria have many sigma factors and the regulons of many these are not well defined)?

* Thank you very much for this important suggestion. We searched for conserved sequence motifs within the regions upstream of the genes downregulated in the $\Delta sipA$ strain using the MEME algorithm. The computational search produced an enriched sequence motif: 5'-CTC-n₁₈-CCGAA-3'. This sequence motif is highly similar to the target promoters of three FliA-family sigma factors in *A. missouriensis* (Hashiguchi *et al.* 2020. *Mol Microbiol* **113**:1170-1188). As shown in the Supplementary Data 3, the most excessively downregulated gene in the $\Delta sipA$ strain is *fliA3* (downregulated 40.8-fold in mutant $\Delta sipA$), which encodes one of the FliA-family sigma factors. Therefore, we supposed that the downregulation of *fliA3* led to lower transcript levels of the FliA3 regulon in the $\Delta sipA$ strain. Although the transcription of *fliA3* is upregulated during sporangium formation and sporangium dehiscence, TcrA, a key transcriptional activator for sporangium formation and sporangium dehiscence, is required for the transcription of *fliA3*, and the principal sigma factor seems to be used for the transcription of *fliA3*. Therefore, downregulation of *fliA3* in the $\Delta sipA$ strain is not due to sigma factor competition between σ^{SsdA} and another alternative sigma factor. Furthermore, because downregulation of the FliA3 regulon seems to be caused by a lower amount of FliA3, it is also not due to sigma factor competition between σ^{SsdA} and FliA3. In addition, it should be noted that *fliA3* has no apparent role in the morphological development including sporangium dehiscence in *A. missouriensis* (Hashiguchi *et al.* 2020. *Mol Microbiol* **113**:1170-1188). Thus, we identified no regulon controlled by an alternative sigma factor among the 333 downregulated genes identified in the RNA-seq experiment.

Reviewer #3

The authors have carefully considered and acted on the reviewers' comments which has made important improvements to the manuscript. Most significantly, the authors have agreed that the sigma/ anti-sigma couple that was identified in a screen for mutants

involved in sporangium dehiscence, might not be involved in this process. Rather sigma competition caused by hyperexpression of the sigma factor could account for the phenotypes.

1. Although the authors agree that the phenotype seen in SsdA mutants might be caused by sigma competition they maintain that the regulatory couple are involved somehow in sporangiospore physiology. This is primarily because the genes are expressed during sporangium formation and dehiscence, although presumably the same could be said for any gene that is expressed during this process. The fact that double mutants are not affected in either process suggests that any physiological role is minimal. Ultimately, even though the title of the paper has changed to reflect the likelihood that the sigma/anti-sigma system is involved in oxidative stress response rather than sporangiogenesis/dehiscence, the paper retains a focus on the latter and therefore is somewhat misleading to a reader looking for insights into this fascinating morphological development process. The study does identify a two-component system that looks likely to be involved in dehiscence, but is not likely to be SsdA/SipA controlled and is only studied briefly.

* Thank you very much for this constructive criticism. We agree with this reviewer that the direct role of the SipA/ σ^{SsdA} system in sporangiospore physiology is not clear, except for oxidative stress tolerance, considering that no phenotypic changes were observed between the wild-type and $\Delta ssdA$ strains. However, we believe that the SipA/ σ^{SsdA} system is involved in the physiological maturation of sporangium and sporangiospore including oxidative stress responses, because the transcription of *sipA*, *ssdA*, and its target genes occurs during sporangium formation and sporangium dehiscence. According to this comment and the editor's comments, we have changed the manuscript title and abstract, and rephrased the claims that the SipA/ σ^{SsdA} system is involved in sporangiospore physiology in the revised manuscript as follows (L. 1-2; L. 19-31; L. 96-99; L. 201-202; L. 567-569; L. 575-580; L. 600-602; L. 693-695). We have removed the claim that this system is related to sporangiospore physiology from the title and abstract, and have toned down it in the main text in general. However, we would ask for your understanding and approval to mention our thoughts as a possibility, especially in the Discussion section.

“A unique sigma/anti-sigma system in the actinomycete *Actinoplanes missouriensis*”

“Bacteria of the genus *Actinoplanes* form sporangia that contain dormant sporangiospores which, upon contact with water, release motile spores (zoospores) through a process called sporangium dehiscence. Here, we set out to study the molecular mechanisms behind sporangium dehiscence in *Actinoplanes missouriensis* and discover a sigma/anti-sigma system with unique features. Protein σ^{SsdA} contains a functional sigma factor domain and an anti-sigma factor antagonist domain, while protein SipA contains an anti-sigma factor domain and an anti-sigma factor antagonist domain. Remarkably, the two proteins interact with each other via the anti-sigma factor antagonist domain of σ^{SsdA} and the anti-sigma factor domain of SipA. Although it remains unclear whether the SipA/ σ^{SsdA} system plays direct roles in sporangium dehiscence, the system seems to modulate oxidative stress responses in zoospores. In addition, we identify a two-component regulatory system (RsdK-RsdR) that represses initiation of sporangium dehiscence.”

“We identified a protein pair composed of the sigma factor σ^{SsdA} (AMIS_54240) and its cognate anti-sigma factor SipA (AMIS_54230), which is involved in oxidative stress responses in zoospores.”

“We designated AMIS_54240 as SsdA (sigma factor presumably involved in sporangium dormancy).”

“In this study, using classical forward genetic methods, we identified the SipA- σ^{SsdA} sigma/anti-sigma system that is involved in oxidative stress responses of sporangiospores in *A. missouriensis*.”

“(I) the presence of a sigma factor that is presumably involved in physiological maturation (not morphological maturation) of sporangium and sporangiospore, including acquisition of oxidative stress resistance, and (II) the presence of a molecular mechanism that delays sporangium dehiscence. The predicted gene regulatory system that is presumably involved in physiological maturation in sporangiospores is schematically represented in Fig. 7.”

“Nevertheless, we believe that σ^{SsdA} is involved in physiological maturation of sporangium and sporangiospore including oxidative stress responses, as described in the following paragraph.”

“We propose that σ^{SsdA} activates its regulon involved in physiological maturation of sporangium and sporangiospore including oxidative stress responses.”

2. The domain organisation of the sigma and anti-sigma couple are certainly interesting and suggests complex regulatory mechanism, the inability to isolate soluble SipA means that it is uncertain how these domains interact and how a signal, such as oxidative stress, is perceived.

* Yes, it is completely unknown how a signal, such as oxidative stress, is perceived by the sigma/anti sigma system. It is also possible that no signals modulate the sigma/anti sigma system; the amounts of both proteins regulate the sigma factor activity. Meanwhile, we experimentally demonstrated that SipA and σ^{SsdA} interact with each other via the interaction between the anti-sigma factor domain of SipA and the anti-sigma factor antagonist domain of σ^{SsdA} using bacterial two-hybrid assays. Furthermore, the structure of the SipA/ σ^{SsdA} complex predicted by AlphaFold-Multimer supported this interaction and provided an important insight into the binding mechanism of these domains from SipA and σ^{SsdA} . We believe that the molecular mechanism underlying the regulation of σ^{SsdA} by SipA is an important future challenge.

3. The paper now shows that SsdA/SipA is responsible for oxidative stress resistance in zoospores, however it is not mentioned whether the system is important during vegetative growth. If so, this would argue for a more general role of the system in oxidative stress rather than a specific role in sporangiogenesis.

* Thank you very much for this important suggestion. Unfortunately, we have no experimental data on the role of the SipA/ σ^{SsdA} system during vegetative growth. In addition, it is difficult to examine the resistance of vegetative hyphae (which form mycelia) to H_2O_2 by measuring colony-forming units. However, we assume that the SipA/ σ^{SsdA} system is unlikely to be involved in oxidative stress responses during vegetative growth, considering that the transcript levels of *sipA*, *ssdA*, and σ^{SsdA} -dependent genes are very low during vegetative growth (Fig. S8). We have added the following description in the revised manuscript (L. 559-564).

“We assume that σ^{SsdA} is unlikely to be involved in the oxidative stress response during vegetative growth because the transcript levels of *ssdA* and σ^{SsdA} -dependent genes are

very low during vegetative growth (Fig. S8). Thus, these results support our assumption that the function of σ^{SsdA} is analogous to that of the stationary phase-specific sigma factor σ^{S} .”